# dMARK: Decoding-Guided Watermarking for Discrete Diffusion Language Models

## Abstract

We introduce dMARK, the first decoding-guided watermarking method for discrete diffusion language models (dLLMs). Unlike prior approaches that modify token probabilities, dMARK embeds watermark signals by steering the decoding order according to a binary hashing rule that prioritizes tokens whose indices match a target parity, leaving the underlying probability distribution intact. dMARK is broadly compatible with common decoding strategies (e.g., confidence, entropy, and margin-based) and can be further enhanced with beam search. Experiments on multiple dLLMs and benchmark datasets show that dMARK achieves strong detectability with minimal quality degradation. The watermark also remains robust under post-editing operations, including insertion, deletion, substitution, and paraphrasing, establishing decoding-guided watermarking as a practical solution for dLLMs.

## 1 Introduction

Large Language Models (LLMs) have achieved remarkable progress in generating coherent and high-quality text, enabling applications in question answering (Yue, 2025), programming (Jiang et al., 2024), and academic writing (Perkins, 2023). At the same time, the ability of LLMs to generate human-like text poses serious risks. Machine-generated content can be weaponized for disinformation (Ranade et al., 2021), phishing (Karanjai, 2022), or plagiarism (Kasneci et al., 2023), and can exacerbate issues of copyright infringement (Rillig et al., 2023), identity theft (Kumar et al., 2024), and fraud (Mirsky et al., 2023). As LLMs become more accessible, reliable methods to distinguish machine-generated from human-authored text are urgently needed (Bender et al., 2021; Crothers et al., 2023).

Watermarking has emerged as one of the most practical approaches for content provenance. In LLMs, watermarking methods embed imperceptible statistical signals that can later be detected (Feng et al., 2025; Wu et al., 2025). Existing approaches, however, are largely designed for autoregressive models (ARMs) and fall into two categories. The first biases token probabilities, typically by partitioning the vocabulary into "green" and "red" lists (Kirchenbauer et al., 2023; Zhao et al., 2024); while effective in theory, this biasing distorts the output distribution and degrades text quality. The second aims for distortion-free watermarking, where token probabilities remain unchanged but token selection is conditioned on long pseudo-random key sequences (Kuditipudi et al., 2024); although this avoids distributional shifts, it requires long keys, which slow detection and limit scalability. Despite their differences, both approaches fundamentally assume left-to-right generation, restricting applicability to ARMs.

Recent works have introduced *discrete diffusion language models* (dLLMs) (Lou et al., 2024; Nie et al., 2025) as a promising alternative to the widely deployed autoregressive paradigm. By modeling conditional probabilities under arbitrary masking patterns, dLLMs match or exceed the performance of ARMs in low-resource regimes, while offering additional benefits such as adaptive decoding and controllable generation (Yu et al., 2025; Li et al., 2025). Despite the recent progress of dLLMs, a watermarking method specifically tailored to these models remains underexplored. This gap motivates our work on decoding-based watermarking designed for their order-agnostic generation process.

This paper introduces **dMARK**, the first watermarking method designed for dLLMs. Rather than modifying token probabilities, dMARK embeds a watermark by adjusting *which position is revealed first*. At each decoding step, a hashing function assigns every candidate token a binary value, and

Figure 1: **Overview.** **(Left)** Existing autoregressive watermarking methods generate green/red token sets by hashing the preceding context and embed watermark signals by biasing the sampling distribution toward green tokens. **(Middle)** In contrast, decoding in dLLMs does not follow the traditional left-to-right generation process; instead, the model selects high-reward tokens at each position even in the absence of prior context. **(Right)** The proposed method leverages these rewards and embeds watermark signals by prioritizing tokens with high reward that satisfy the parity condition.

the decoder prioritizes positions where the binary value matches the parity of the position index. Over an entire sequence, this creates a systematic bias: the proportion of tokens aligned with the parity condition (the matching ratio) is systematically greater than the $0.5$ baseline expected under randomness. This ratio then serves as the test statistic for watermark detection, while the underlying distribution $p_\theta(y|x)$ remains unchanged, preserving text quality.

This design has important implications. First, dMARK is *general-purpose*: it can be combined with any decoding strategy, including confidence, entropy, margin, or greedy-based rules. Second, it introduces no distortion to the output distribution, providing quality preservation superior to probability-biasing approaches. Third, it is inherently *robust*: even after tokens are inserted, deleted, or substituted, watermark traces persist. By combining parity-guided decoding with a sliding-window detection procedure, dMARK reliably identifies watermarked text, even under heavy random edits or paraphrasing, because shifts in token alignment produce predictable deviations from the baseline matching ratio of $0.5$.

We evaluate dMARK on benchmark datasets using state-of-the-art dLLMs, including LLaDA and Dream. Our experiments show that dMARK achieves strong detectability, minimal quality loss, and resilience against a wide range of post-editing scenarios. These results demonstrate that decoding-guided watermarking is not only feasible but also a practical path toward reliable provenance in order-agnostic language models.

## 2 RELATED WORKS

**Watermarking in LLMs.** Digital watermarking has long been used to trace provenance and embed imperceptible signals across text, images, and other media (Petitcolas et al., 1999; Zhu et al., 2018; Liang et al., 2024). In LLMs, most methods embed watermarks by *biasing token probabilities*. A common approach (Kirchenbauer et al., 2023; Zhao et al., 2023; 2024) partitions the vocabulary into "green" and "red" sets, increasing the probability of green tokens and detecting watermarks via statistical tests. While theoretically grounded, such biasing can distort generation quality. Distortion-free variants (Kuditipudi et al., 2024; Christ et al., 2024) preserve probability distributions but require long keys and remain tailored to left-to-right generation.

More recently, Chen et al. (2025) extended watermarking to order-agnostic models, but their method still biases token probabilities, preserving the trade-off between detectability and text quality. In contrast, our work is the first to exploit the *decoding strategy* of discrete diffusion language models (dLLMs), embedding watermarks without modifying probabilities and ensuring compatibility with diverse decoding strategies.

**Discrete Diffusion LLMs.** Diffusion models (Ho et al., 2020; Song et al., 2021; 2022) have achieved strong results in continuous domains such as images (Rombach et al., 2022; Saharia et al., 2022) and have been adapted to discrete domains through Masked Diffusion Models (MDMs) (Austin et al., 2021; Lou et al., 2024; Sahoo et al., 2024; Shi et al., 2024; Ou et al., 2025), which iteratively denoise masked tokens. A key property of MDMs, and by extension dLLMs, is their *order-agnostic generation*: they learn to model conditional distributions under arbitrary masking patterns. This order-agnostic nature admits a wide variety of decoding strategies (e.g., random,

| **Algorithm 1** Generic dLLMs Decoding | **Algorithm 2** dMARK: Watermarks by Decoding |
|---|---|
| **Require:** Prompt $x$; output length $n$; predictor $p_\theta$; decoding strategy $\mathcal{F}$ | **Require:** Prompt $x$; output length $n$; predictor $p_\theta$; decoding strategy $\mathcal{F}$; matching set $\mathcal{G}_j$ |
| 1: $y \leftarrow [\text{MASK}]^n$; $\quad \mathcal{I} \leftarrow \emptyset$ | 1: $y \leftarrow [\text{MASK}]^n$; $\quad \mathcal{I} \leftarrow \emptyset$ |
| 2: **for** $i = 1, \dots, n$ **do** | 2: **for** $i = 1, \dots, n$ **do** |
| 3: $\quad$ Get $\{(r_j, v_j) = \mathcal{F}(j; p_\theta, x, y_{\mathcal{I}}) | j \notin \mathcal{I}\}$ | 3: $\quad$ Get $\{(r_j, v_j) = \mathcal{F}(j; p_\theta, x, y_{\mathcal{I}}) \mid j \notin \mathcal{I}\}$ |
| 4: $\quad \mathcal{C} \leftarrow \{j \notin \mathcal{I}\}$ | 4: $\quad \mathcal{C} \leftarrow \{j \notin \mathcal{I} \mid v_j \in \mathcal{G}_j\}$ |
| 5: $\quad k^\star \leftarrow \arg\max_{j \in \mathcal{C}} r_j$ | 5: $\quad$ **if** $\mathcal{C} = \emptyset$ **then** $\mathcal{C} \leftarrow \{j \notin \mathcal{I}\}$ **endif** |
| 6: $\quad y_{k^\star} \leftarrow v_{k^\star}$; $\quad \mathcal{I} \leftarrow \mathcal{I} \cup \{k^\star\}$ | 6: $\quad k^\star \leftarrow \arg\max_{j \in \mathcal{C}} r_j$ |
| 7: **end for** | 7: $\quad y_{k^\star} \leftarrow v_{k^\star}$; $\quad \mathcal{I} \leftarrow \mathcal{I} \cup \{k^\star\}$ |
| 8: **Return** $y$ | 8: **end for** |
| | 9: **Return** $y$ |

confidence, entropy, and margin-based), making dLLMs more flexible than autoregressive models (ARMs), where generation is strictly left-to-right.

Recent large-scale dLLMs such as LLaDA (Nie et al., 2025; Zhu et al., 2025) and Dream (Ye et al., 2025) demonstrate that this framework scales competitively, often matching or surpassing autoregressive models (ARMs), especially in low-resource regimes (Prabhudesai et al., 2025). Industrial systems including Mercury (Labs et al., 2025) and Gemini Diffusion (DeepMind, 2025) further showcase the efficiency of dLLMs. These advances motivate watermarking methods specifically tailored to dLLMs. Our work addresses this gap by introducing the first decoding-guided watermarking scheme, exploiting their order-agnostic property and diverse decoding strategies to embed robust watermarks without sacrificing quality.

## 3 DECODING-GUIDED WATERMARKING FOR dLLMs

### 3.1 GENERIC DECODING STRATEGY

Discrete diffusion language models (dLLMs) differ from autoregressive models in a crucial way: instead of being forced to generate text strictly from left to right, they can in principle reveal tokens in *any order*. This property arises because dLLMs are trained to predict a missing token given an arbitrary subset of revealed tokens. As a result, the same sequence can be generated through many different decoding orders, making decoding strategy an essential design choice.

Formally, let $p_{\text{data}}$ denote the true data distribution. Given a prompt $x = (x_1, \dots, x_m)$, the goal of dLLMs is to generate a sequence $y = (y_1, \dots, y_n)$ such that $y \sim p_{\text{data}}(y|x)$. For any subset of revealed indices $\mathcal{I} \subset \{1, \dots, n\}$ (where $y_j$ is revealed for $j \in \mathcal{I}$) and a target index $i \notin \mathcal{I}$, the dLLMs learn a predictor $p_\theta$ that approximates

$$p_\theta(y_i|y_{\mathcal{I}}, x) \approx p_{\text{data}}(y_i|y_{\mathcal{I}}, x),$$

while treating the remaining tokens as $[\text{MASK}]$. This means that, ideally, the distribution of $y$ can be factorized along any permutation $\pi$ of $\{1, \dots, n\}$:

$$p_{\text{data}}(y|x) = \prod_{i=1}^{n} p_{\text{data}}(y_{\pi(i)}|y_{\pi(<i)}, x) \approx \prod_{i=1}^{n} p_\theta(y_{\pi(i)}|y_{\pi(<i)}, x),$$

where $y_{\pi(<i)} = \{y_{\pi(k)}|k < i\}$. In theory, the choice of order $\pi$ should not matter. In practice, however, imperfect training causes different decoding strategies to yield different results, making the decoding strategy a central component of dLLMs generation (Kim et al., 2025). Accordingly, the watermark signal observed in practice arises from approximation error and decoding heuristics.

At each decoding step $i$, let $\mathcal{I} = \{\pi(1), \dots, \pi(i-1)\}$ be the set of revealed indices. A decoding strategy $\mathcal{F}(j; p_\theta, x, y_{\mathcal{I}})$ returns, for each unrevealed index $j \notin \mathcal{I}$, a reward $r_j$ and a sampled candidate token $v_j$. The next index is then chosen as $\pi(i) = \arg\max_{j \notin \mathcal{I}} r_j$, and the corresponding token will be $y_{\pi(i)} \leftarrow v_{\pi(i)}$. This generic decoding procedure is summarized in Algorithm 1.

A range of decoding strategies $\mathcal{F}$ have been proposed, reflecting trade-offs between certainty and exploration (Nie et al., 2025; Ye et al., 2025; Kim et al., 2025). Common examples include:

- **Random**: rewards $r_j$ are sampled uniformly at random; sample $v_j \sim p_\theta(y_j = \cdot | y_\mathcal{I}, x)$.
- **Confidence**: sample $v_j \sim p_\theta(y_j = \cdot | y_\mathcal{I}, x)$ and set $r_j = p_\theta(y_j = v_j | y_\mathcal{I}, x)$.
- **Entropy**: set $r_j = -H(Y_j \mid y_\mathcal{I}, x)$, where $H(\cdot)$ denotes the conditional entropy under $p_\theta$.
- **Margin**: let $v_j = \arg\max_v p_\theta(v \mid y_\mathcal{I}, x)$ and define $r_j$ as the probability gap between the top-1 and top-2 candidates.

In decoding strategies that involve stochastic sampling, the token $v_j$ may either be drawn from $p_\theta(\cdot \mid y_\mathcal{I}, x)$ or chosen greedily as

$$v_j = \arg\max_v p_\theta(y_j = v | y_\mathcal{I}, x).$$

Finally, although parallel decoding methods exist that reveal multiple tokens at once to speed up generation (Ben-Hamu et al., 2025; Wei et al., 2025), here we focus on the sequential framework. This provides a clean foundation on which we will build our watermarking method.

## 3.2 PROBLEM SETUP

Our goal is to design a watermarking strategy specifically for dLLMs. In this setting, watermarking means that a sequence $y \sim p_{\text{data}}(y|x)$ and a sequence $y' \sim p_\theta(y|x)$ generated by dLLMs can be made *statistically distinguishable*, while still preserving the performance of the model.

Most existing watermarking methods for LLMs achieve detectability by biasing token probabilities. Although effective, such approaches distort the model's output distribution and can noticeably degrade text quality. dLLMs, however, offer a unique opportunity: because $p_\theta(y|x)$ is, in principle, invariant to the order in which tokens are generated, watermarking can instead be realized by modifying the decoding strategy $\mathcal{F}$ rather than altering probabilities.

In our formulation, the watermark is embedded directly through the decoding process. We design an *adaptive ordering strategy* in which the choice of which position to unmask is guided to enhance detectability. This embeds a signal without altering the generating distributions, ensuring that the watermark remains imperceptible to humans and minimally invasive to the model's outputs.

We formalize watermarking for dLLMs as a decoding problem subject to three requirements:

1. **Performance preservation**: The decoding procedure must not distort the underlying distribution $p_\theta(y|x)$, ensuring that text quality is maintained.
2. **Detectability**: The watermark must be verifiable from the generated text $y'$ and the watermark key alone, without requiring access to model internals or prompts.
3. **Robustness**: The watermark must remain detectable even under random or adversarial modifications of the text (e.g., insertions, deletions, and substitutions).

## 3.3 dMARK: DECODING-GUIDED WATERMARKING FOR dLLMS

We now present **dMARK**, our watermarking method for dLLMs. The approach is inspired by prior LLM watermarking schemes (Kirchenbauer et al., 2023), which conceptually divide the vocabulary into two groups and analyze the frequency of designated tokens. In contrast to those methods, which modify token probabilities, dMARK embeds the watermark by guiding the decoding order. A lightweight binary hashing rule determines which candidate tokens align with the position index, and the decoder simply prioritizes those positions. This embeds a detectable signal while leaving the conditional probabilities $p_\theta(y_j | y_\mathcal{I}, x)$ unchanged, thereby preserving generation quality.

Given a watermark key $\xi$, we define a deterministic hashing function $f : \mathcal{V} \times \Xi \to \{0, 1\}$ that maps each token $v \in \mathcal{V}$ to a binary value conditioned on $\xi$. The function is constructed so that, for any key $\xi$, the resulting partition is balanced. At each position $i$, the vocabulary is divided as

$$\mathcal{G}_i = \{v \in \mathcal{V} \mid f(v, \xi) \equiv i \pmod 2\}, \qquad \mathcal{R}_i = \mathcal{V} \setminus \mathcal{G}_i,$$

where $\mathcal{G}_i$ is the parity-matching set and $\mathcal{R}_i$ is the residual set.

During decoding, indices whose predicted tokens fall in $\mathcal{G}_i$ are prioritized; that is, we select the index with the largest reward among parity-matching candidates. If no such index exists, the procedure falls back to $\mathcal{R}_i$. This strategy is summarized in Algorithm 2.

**Prompt** *The most saddest love story you can ever think of* 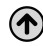

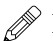 **Non-watermarked**

In a small village, there lived a young woman named Lily. She was a kind-hearted and gentle soul, always willing to help others. One day, she met a young man named Jack, who was also kind-hearted and gentle. They fell in love and spent their days together, sharing laughter and tears.

However, their happiness was short-lived. One day, a terrible earthquake struck the village, killing many people, including Lily and Jack. The village was left in ruins, and the survivors were left to pick up the pieces of their lives. [...continues]

··· **Parity-Matching Ratio = 0.55**

**Watermarked (dMARK)**

In a small town, there was a young girl named Lily who lived in a small house with her parents. One day, she met a young man named Jack who was from a wealthy family. They shared a lot in common and soon fell deeply in love.

However, their happy moment was short-lived. Lily's parents did not approve of their relationship, and they forced Jack to leave Lily behind. Jack was heartbroken, but he did not give up on her. He waited for Lily for many years, but she never came back. [...continues]

··· **Parity-Matching Ratio = 0.74**

Figure 2: **Non-watermarked vs. watermarked text.** Generated by LLaDA-1.5 on the Writing Prompts dataset. Tokens highlighted in green indicate parity matches.

The dMARK framework is compatible with any decoding strategy $\mathcal{F}$. Within dMARK, the deocding order $\pi$ is adjusted so that positions with parity-matching candidates are filled first, while remaining positions are handled afterward. The resulting text is indistinguishable from standard decoding, yet statistically it exhibits a systematic bias toward parity-matching positions, which provides a reliable watermarking signal.

### 3.4 dMARK WITH BEAM SEARCH

The standard version of dMARK selects the index with the highest reward among those whose predicted tokens belong to $\mathcal{G}_i$. Although straightforward, this choice can commit too early to a local optimum, which may reduce the number of positions satisfying the parity constraint in later steps.

To address this limitation, we propose a generalized beam-search variant with one-step lookahead. At each step, the algorithm first identifies the top-$k$ indices $\mathcal{T}$ with the largest rewards, as in standard dMARK. For each candidate $j \in \mathcal{T}$, a lookahead score is then computed to estimate how many future positions will remain parity-consistent if we commit to $(j, v_j)$.

Formally, given the revealed set $\mathcal{I}$, the lookahead score for candidate $j$ is defined as

$$g^{(j)} = \sum_{\ell \notin \mathcal{I} \cup \{j\}} \mathbb{1}\left[\hat{v}_\ell \in \mathcal{G}_\ell\right],$$

where $\hat{v}_\ell$ denotes the greedy prediction from $p_\theta$ at position $\ell$ after committing $y_j \leftarrow v_j$. This score approximates the number of remaining positions expected to satisfy the parity condition in the next step. The selected index is then chosen as $k^\star = \arg\max_{j \in \mathcal{T}} g^{(j)}$.

This balances immediate reward maximization with preserving parity consistency across future steps. When $k = 1$, the method reduces exactly to the greedy dMARK strategy. For larger $k$, the decoder trades off efficiency for greater robustness, as the lookahead mechanism preserves more opportunities for parity alignment in later decoding steps.

### 3.5 WATERMARK DETECTION

We now describe how to detect the presence of a watermark in generated text. Recall that our scheme partitions the vocabulary at each position $i$ into a parity-matching set $\mathcal{G}_i$ and a residual set $\mathcal{R}_i$. Given access to $(f, \xi)$, verification can be performed directly on the generated sequence without requiring the prompt or access to model internals.

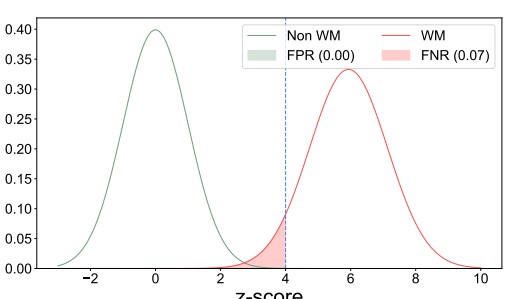

**Figure 3: Illustration of z-scores computed from matching ratios of non-watermarked (Non WM) and watermarked (WM) texts.** The false negative rate (FNR) at a threshold of 4 is highlighted in red, while the false positive rate (FPR) is measured as 0.

**Table 1: Empirical error rates for watermark detectability on the C4 and Writing Prompts datasets.** The results are based on texts generated by multiple dLLMs with multinomial sampling at a $z$-score threshold of 4.

| Dataset | Model | PPL | $z = 4.0$ | |
|---|---|---|---|---|
| | | | FPR | FNR |
| C4 | LLaDA | 4.90 | 0.000 | 0.043 |
| | LLaDA 1.5 | 5.27 | 0.000 | 0.071 |
| | Dream | 5.75 | 0.000 | 0.042 |
| Writing Prompts | LLaDA | 6.00 | 0.000 | 0.017 |
| | LLaDA 1.5 | 6.34 | 0.004 | 0.267 |
| | Dream | 6.87 | 0.007 | 0.318 |

**Basic Detection.** Given a sequence of $n$ tokens, each token $y_i$ is checked for membership in $\mathcal{G}_i$, and let $G$ denote the number of matches. For non-watermarked text, token assignments to $\mathcal{G}_i$ are effectively random, so the expected matching ratio converges to $\gamma = \frac{1}{2}$. In contrast, dMARK produces a systematically higher matching ratio, since the decoding order explicitly prioritizes parity-matching indices. This deviation can be quantified using a one-sided $z$-test:

$$z = \frac{G/n - \gamma}{\sqrt{\gamma(1-\gamma)/n}}, \tag{1}$$

and the sequence is flagged as watermarked if $z$ exceeds a predefined threshold.

**Robust Detection.** While basic detection is effective for clean generations, post-editing operations such as insertions, deletions, or substitutions may disrupt parity alignment. In particular, insertions or deletions induce a *parity shift*, flipping the alignment of subsequent tokens. As a result, the matching ratio beyond the shift often drops below $\frac{1}{2}$, producing inverted signals.

To address this, we adopt a sliding-window detection strategy. The sequence is divided into overlapping windows of length $w$, and the local matching ratio is computed within each window. For non-watermarked text, window-level ratios remain concentrated around $\frac{1}{2}$. For watermarked text, they cluster around $\alpha > \frac{1}{2}$. When edits occur, the distribution becomes multimodal: some windows remain aligned with the watermark (peaking near $\alpha$), while others flip after a parity shift (peaking near $1 - \alpha$). This multimodality provides a clear indicator of watermark (see Appendix E, Figure 7).

Formally, detection is performed by computing the $z$-score in Equation (1) for each window and aggregating the results across windows. A sequence is classified as watermarked if a fraction of windows exhibit matching ratios that deviate from $\frac{1}{2}$. This sliding-window approach preserves detectability even under extensive edits, providing robustness to insertion, deletion, and substitution.

## 4 EXPERIMENTAL EVALUATION

### 4.1 EXPERIMENTAL SETUP

**Datasets and Prompts.** We use two benchmark datasets. The first is the news-like subset of C4 (Raffel et al., 2023), which has been widely employed in prior watermarking studies (Kirchenbauer et al., 2023; Kuditipudi et al., 2024; Block et al., 2025; Feng et al., 2025). The second is Writing Prompts (Fan et al., 2018), which provides diverse topics and narrative styles, ranging from apocalyptic scenarios to everyday stories. For C4, we randomly sample texts and truncate them to a fixed length to serve as prompts; for Writing Prompts, the given prompts are used directly.

**Models and Environments.** Experiments are conducted on LLaDA-8B (Instruct) (Nie et al., 2025), LLaDA 1.5-8B (also instruction-tuned) (Zhu et al., 2025), and Dream-7B (Instruct) (Ye et al., 2025). All models generate sequences of length 256 using block-wise generation (Arriola

**Table 2: Watermark Detectability Comparison.** Empirical results under greedy and multinomial sampling with LLaDA 1.5 on the C4 dataset, reported across two evaluation metrics: fixed $z = 4.0$ and TPR@FPR. Greedy and multinomial sampling represent the non-watermarked baselines.

| Method | PPL ↓ | z = 4.0 | | | | TPR@FPR ↑ | | | |
| | | FPR↓ | TNR↑ | TPR↑ | FNR↓ | 10% | 1% | 0.1% | 0.01% |
|---|---|---|---|---|---|---|---|---|---|
| **Greedy Sampling** | 4.03 | - | - | - | - | - | - | - | - |
| KGW ($\delta = 1$) | 4.33 | 0.0 | 1.0 | 0.072 | 0.928 | 88.52 | 62.68 | 30.14 | 11.48 |
| KGW ($\delta = 2$) | 5.02 | 0.0 | 1.0 | 0.866 | 0.134 | 100.00 | 97.31 | 93.01 | 97.63 |
| KGW ($\delta = 3$) | 5.83 | 0.0 | 1.0 | 0.970 | 0.030 | 100.00 | 100.00 | 98.52 | 97.78 |
| PATTERN-MARK ($\delta = 1$) | 4.11 | 0.0 | 1.0 | 0.000 | 1.000 | 21.76 | 4.17 | 1.39 | 0.00 |
| PATTERN-MARK ($\delta = 2$) | 4.72 | 0.0 | 1.0 | 0.040 | 0.960 | 73.50 | 48.50 | 20.50 | 12.00 |
| PATTERN-MARK ($\delta = 3$) | 5.86 | 0.0 | 1.0 | 0.584 | 0.416 | 96.26 | 91.59 | 87.38 | 78.97 |
| dMARK | **4.44** | 0.0 | 1.0 | 0.540 | 0.460 | 97.86 | 91.98 | 76.47 | 60.96 |
| + 3-beam | 4.75 | 0.0 | 1.0 | 0.963 | 0.037 | 100.00 | 99.54 | 98.62 | 97.25 |
| + 5-beam | 5.01 | 0.0 | 1.0 | 0.987 | 0.013 | 100.00 | 100.00 | 99.56 | 98.69 |
| + 8-beam | 5.16 | 0.0 | 1.0 | 0.991 | 0.008 | 100.00 | 100.00 | 100.00 | 99.12 |
| **Multinomial Sampling** | 4.21 | - | - | - | - | - | - | - | - |
| KGW ($\delta = 1$) | 5.59 | 0.0 | 1.0 | 0.107 | 0.893 | 89.80 | 60.91 | 32.99 | 14.21 |
| KGW ($\delta = 2$) | 6.38 | 0.0 | 1.0 | 0.876 | 0.124 | 99.41 | 98.82 | 97.65 | 91.18 |
| KGW ($\delta = 3$) | 7.87 | 0.0 | 1.0 | 0.984 | 0.016 | 100.00 | 99.21 | 99.21 | 98.41 |
| PATTERN-MARK ($\delta = 1$) | 5.45 | 0.0 | 1.0 | 0.000 | 1.000 | 25.26 | 5.67 | 1.55 | 0.00 |
| PATTERN-MARK ($\delta = 2$) | 6.33 | 0.0 | 1.0 | 0.060 | 0.940 | 78.00 | 53.50 | 27.50 | 16.50 |
| PATTERN-MARK ($\delta = 3$) | 7.69 | 0.0 | 1.0 | 0.586 | 0.414 | 98.99 | 95.96 | 91.41 | 83.33 |
| dMARK | **5.27** | 0.0 | 1.0 | 0.929 | 0.071 | 100.00 | 100.00 | 99.41 | 95.29 |
| + 3-beam | 5.40 | 0.0 | 1.0 | 1.000 | 0.000 | 100.00 | 100.00 | 100.00 | 100.00 |
| + 5-beam | 5.76 | 0.0 | 1.0 | 1.000 | 0.000 | 100.00 | 100.00 | 100.00 | 100.00 |
| + 8-beam | 6.00 | 0.0 | 1.0 | 1.000 | 0.000 | 100.00 | 100.00 | 100.00 | 100.00 |

**Table 3: Benchmark results for dMARK.** Evaluated on LLaDA, LLaDA 1.5, and Dream under greedy and multinomial sampling. The comparison includes (1) non-watermarked baseline, (2) KGW, (3) PATTERN-MARK, (4) dMARK, and (5) dMARK with 3-beam search.

| Model | Method | Greedy Sampling | | | Multinomial Sampling | | |
| | | MMLU (Acc ↑) | GSM8K (Acc ↑) | HumanEval (Pass@1 ↑) | MMLU (Acc ↑) | GSM8K (Acc ↑) | HumanEval (Pass@1 ↑) |
|---|---|---|---|---|---|---|---|
| | Non-watermarked | 0.648 | 0.797 | 0.427 | 0.594 | 0.775 | 0.360 |
| LLaDA | KGW | 0.558 | 0.662 | 0.092 | 0.520 | 0.464 | 0.055 |
| | PATTERN-MARK | 0.570 | 0.635 | 0.134 | 0.532 | 0.438 | 0.073 |
| | dMARK | **0.647** | **0.787** | **0.280** | **0.588** | **0.735** | **0.226** |
| | dMARK +3-beam | **0.647** | 0.771 | 0.268 | 0.580 | 0.678 | 0.152 |
| | Non-watermarked | 0.650 | 0.821 | 0.400 | 0.601 | 0.808 | 0.348 |
| LLaDA 1.5 | KGW | 0.567 | 0.726 | 0.104 | 0.536 | 0.582 | 0.092 |
| | PATTERN-MARK | 0.579 | 0.670 | 0.152 | 0.540 | 0.513 | 0.079 |
| | dMARK | **0.649** | **0.814** | **0.317** | **0.596** | **0.759** | **0.201** |
| | dMARK +3-beam | **0.649** | 0.774 | 0.207 | 0.588 | 0.723 | 0.134 |
| | Non-watermarked | 0.700 | 0.800 | 0.427 | 0.630 | 0.789 | 0.420 |
| Dream | KGW | 0.558 | 0.661 | 0.287 | 0.523 | 0.444 | 0.134 |
| | PATTERN-MARK | 0.594 | 0.652 | 0.335 | 0.551 | 0.639 | 0.287 |
| | dMARK | **0.695** | **0.746** | **0.470** | **0.647** | **0.686** | **0.390** |
| | dMARK +3-beam | **0.695** | 0.701 | 0.342 | 0.636 | 0.648 | 0.262 |

et al., 2025; Nie et al., 2025). We adopt block sizes of 32 for the LLaDA family and 8 for Dream-7B to encourage longer outputs. Responses are generated for 300 prompts, and we retain sequences longer than 200 tokens (100 tokens for Dream-7B, due to its shorter generations). Text quality is evaluated using perplexity (PPL) computed with Gemma3-12B (Team et al., 2025), a larger model serving as an oracle.

**Sampling Schemes.** We consider two sampling schemes: multinomial sampling, where tokens are drawn from $p_\theta$, and greedy sampling, where the most likely token is chosen at each step. In both settings, the decoding strategy $\mathcal{F}$ follows the confidence rule unless otherwise stated. Beam search, as described in Section 3.4, augments these schemes with one-step lookahead. Additional experiments with entropy and margin-based decoding strategies are reported in the Appendix E.

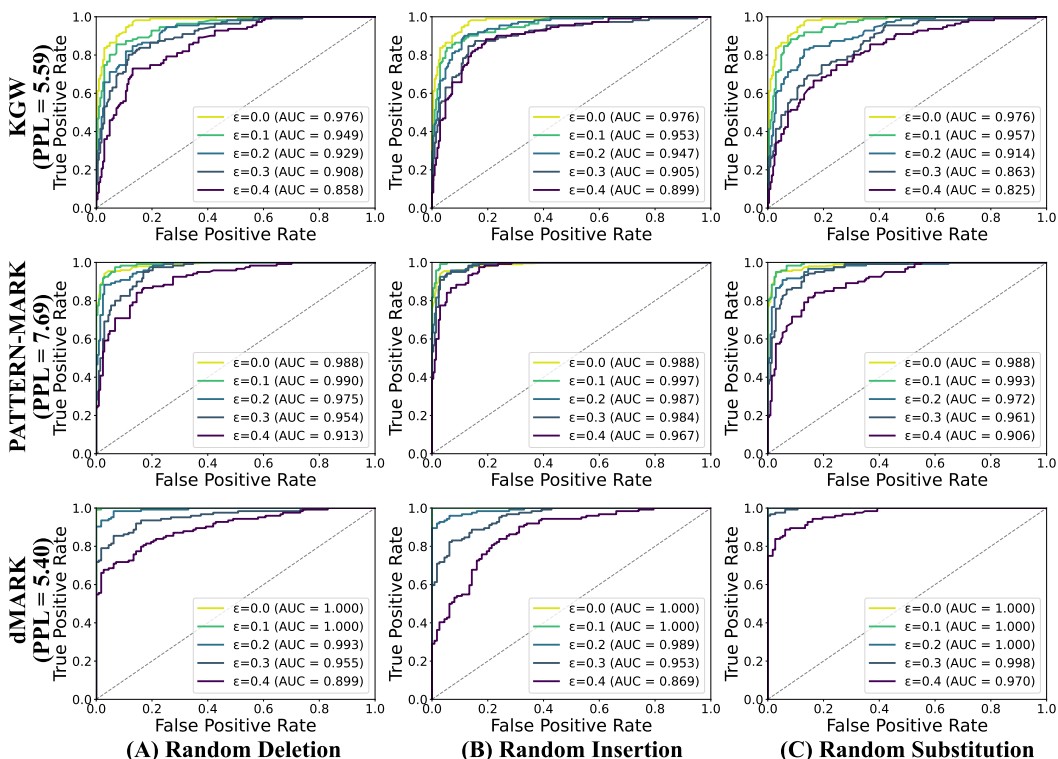

**Figure 4: ROC curves under post-editing attacks.** Illustration of the sliding-window strategy against (A) random deletion, (B) insertion, and (C) substitution with modification budget $\epsilon$. The comparison includes (1) KGW, (2) PATTERN-MARK, and (3) dMARK with 3-beam search.

**Evaluation Metrics.** Performance is assessed along three axes: (1) Detectability: measured by $z$-score (Equation 1), false positive rate (FPR), false negative rate (FNR), and true positive rate at a fixed false positive rate (TPR@FPR). (2) Text Quality: measured by perplexity (PPL) and benchmark accuracy. We include three representative benchmarks using the lm-evaluation-harness (Gao et al., 2023) to measure downstream capability: MMLU (Hendrycks et al., 2021) (multi-task reasoning), GSM8K (Cobbe et al., 2021) (mathematical problem solving), and HumanEval (Chen et al., 2021) (code generation). (3) Robustness: measured by ROC curves under token-level perturbations and paraphrasing attacks.

## 4.2 EXPERIMENTAL ANALYSES

**Watermark Detectability.** Figure 3 shows that watermarked sequences consistently achieve higher $z$-scores than non-watermarked ones, allowing clear separation at a threshold of $z = 4$. Table 1 reports empirical FPR/FNR rates and perplexity (PPL) across LLaDA, LLaDA 1.5, and Dream models on both datasets. These results confirm that dMARK provides reliable detection with negligible error rates while maintaining text quality close to that of non-watermarked text. Illustrative examples in Figure 2 highlight the difference in parity-matching ratios between watermarked and non-watermarked text. Table 2 compares the watermark detectability of dMARK with existing methods, KGW (Kirchenbauer et al., 2023) and PATTERN-MARK (Chen et al., 2025). As KGW employs an autoregressive text generation process, results are derived from dLLMs configured to generate tokens sequentially from left to right. The results indicate that dMARK with 3-beam search consistently achieves higher detectability while maintaining lower PPL compared to existing watermarking methods.

**Effect of Sampling Schemes.** Table 2 compares greedy sampling and multinomial sampling with and without beam search. As beam size increases ($k \in \{1, 3, 5, 8\}$), error rates consistently decrease, demonstrating that one-step lookahead strengthens parity alignment. Multinomial sampling generally yields higher perplexity but produces stronger watermark signals than greedy sampling.

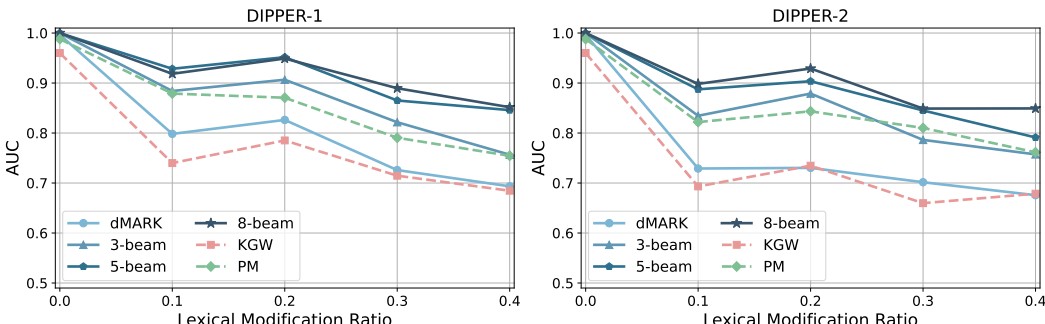

**Figure 5: Detection AUC under paraphrasing attacks.** Results for dMARK with DIPPER (Krishna et al., 2023): (Left) paraphrasing at predefined ratios via lexical modification; (Right) paraphrasing with ratio-adjusted lexical modification and an additional 10% order diversity. Comparative results with KGW and PATTERN-MARK are included to assess relative robustness.

**Text Generation Quality.** Table 3 presents benchmark results on MMLU, GSM8K, and HumanEval under greedy and multinomial sampling, each including results for (1) the non-watermarked baseline, (2) KGW, (3) PATTERN-MARK, (4) dMARK, and (5) dMARK with 3-beam search. We employ $\gamma = 0.5, \delta = 3$ for KGW and PATTERN-MARK, which are chosen to achieve detectability comparable to dMARK with 3-beam search, based on the results in Table 2. On MMLU and GSM8K, watermarking causes only minor degradation, while HumanEval shows a larger drop, consistent with the low entropy of code generation tasks (Lee et al., 2024). Compared to existing methods, dMARK shows minimal degradation in text quality. Table 2 compares perplexity across non-watermarked text, KGW, PATTERN-MARK, and dMARK. Although watermarking increases perplexity relative to the baseline, the increase is negligible compared to probability-biasing methods, since dMARK modifies only the decoding order rather than the token probabilities.

**Detectability-Quality Trade-off.** A clear trade-off emerges between detectability and text quality. As shown in Table 2, larger beam sizes consistently strengthen watermark detectability but also raise perplexity slightly. In practice, we find that $k = 3$ offers a favorable balance, providing strong detection while keeping quality degradation minimal.

**Robustness to Post-editing.** We simulate random token insertions, deletions, and substitutions with perturbation budgets $\epsilon \in \{0.1, 0.2, 0.3, 0.4\}$ and apply sliding-window detection ($w = 8$). Figure 4 presents ROC curves comparing dMARK with existing watermarking schemes, including KGW and PATTERN-MARK. dMARK outputs lower-PPL text while achieving detection performance comparable to existing methods and demonstrating notable robustness to random substitution attacks. Beam search further improves robustness by maintaining better parity alignment. We also evaluate adversarial paraphrasing using DIPPER (Krishna et al., 2023). DIPPER-1 applies paraphrasing at fixed lexical-modification ratios, while DIPPER-2 introduces additional order diversity. As shown in Figure 5, detection performance across KGW, PATTERN-MARK, and dMARK decreases as paraphrasing strength increases. dMARK with 3-beam search remains reliably detectable even at lower PPL compared to existing methods, and larger beam sizes further enhance robustness against paraphrasing.

## 5 PRACTICAL CONSIDERATIONS

The experiments above demonstrated that dMARK achieves strong detectability with minimal quality degradation and remains robust against diverse post-editing attacks. We now analyze two additional factors that influence watermark performance: sequence length and block-wise generation.

**Generation Length.** We evaluate how the number of generated tokens $n$ affects watermark detectability by measuring TPR@FPR ($= 10\%, 1\%, 0.1\%, 0.01\%$) for $n \in \{16, 32, 64, 128, 256\}$. As shown in Figure 6, detection error rates decrease substantially as sequence length increases. When $n \geq 200$, beam search with $k \geq 3$ consistently achieves TPR $= 1.0$ even under a stringent FPR of 0.01. This confirms that watermark reliability improves significantly with longer text.

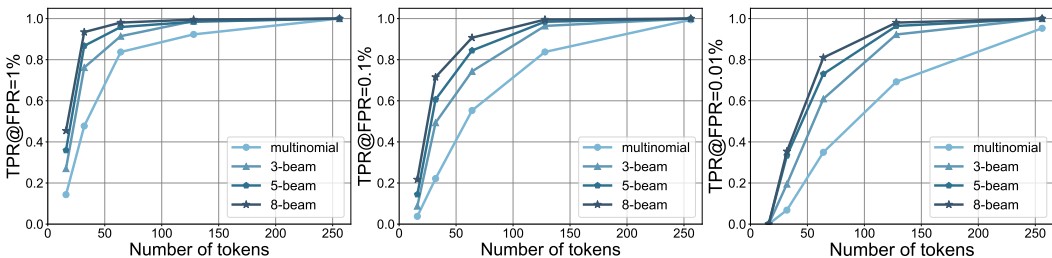

**Figure 6: Watermark detectability vs. sequence length.** Results under multinomial sampling with beam sizes $k \in \{1, 3, 5, 8\}$, reported as TPR at FPR levels of 10%, 1%, 0.1%, and 0.01%.

**Table 4: Effect of block length on watermarking.** TPR at FPR levels of 10%, 1%, 0.1%, and 0.01%, along with PPL on the C4 dataset for dMARK with block lengths of 8, 16, and 32.

| Sampling | Block | PPL | TPR@FPR | | | |
|---|---|---|---|---|---|---|
| | | | 10% | 1% | 0.1% | 0.01% |
| Multinomial | 8 | 4.98 | 100.00 | 97.69 | 93.06 | 80.56 |
| | 16 | 5.16 | 99.50 | 98.02 | 97.03 | 89.60 |
| | 32 | 5.27 | 100.00 | 100.00 | 99.41 | 95.29 |
| Greedy | 8 | 4.28 | 93.42 | 74.12 | 48.25 | 29.82 |
| | 16 | 4.42 | 96.77 | 86.64 | 69.59 | 50.23 |
| | 32 | 4.44 | 97.86 | 91.98 | 76.47 | 60.96 |

**Block-wise Generation.** For efficiency, dLLMs often employ block-wise text generation (Arriola et al., 2025; Nie et al., 2025). To study its effect, we generate sequences of fixed total length (256 tokens) using block sizes $\{8, 16, 32, 64, 128\}$ and report TPR@FPR and PPL in Table 4. Because block sizes of 64 and 128 frequently fail to produce sequences longer than 200 tokens, results are reported only for block sizes up to 32. The Appendix E provides details on the effective sequence lengths under each setting. The results indicate that larger blocks strengthen watermark embedding but that excessively large blocks reduce stability for long-sequence generation.

## 6 CONCLUSION

We introduced dMARK, a decoding-guided watermarking method for discrete diffusion language models (dLLMs). Instead of biasing token probabilities, dMARK embeds watermark signals by guiding the decoding order, preserving the model's original distribution. Comprehensive experiments demonstrate that dMARK provides strong detectability, minimal quality degradation, and robustness against post-editing. These results establish decoding-based watermarking as an effective and practical approach for ensuring provenance in dLLMs.

## ETHICS STATEMENT

This study focuses on embedding watermarks in text generated by dLLMs, and our experiments were conducted using publicly available models and datasets. The study does not involve human participants and contains no elements that pose privacy, legal, or ethical risks. In addition, the work does not produce harmful information or introduce conflicts of interest, discrimination, or bias, and all relevant usage conditions and ethical standards were observed throughout the handling of data.

## REPRODUCIBILITY STATEMENT

To ensure the reproducibility of our findings, we employ publicly available models such as LLaDA-Instruct and Dream-Instruct, and the proposed methodology is described in detail in Section 3. For accessibility and transparency, Table 6 in Appendix D presents the models and datasets used in our experiments, including their references and associated licenses. We plan to release the code used in our experiments to further promote reproducibility.

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

## A   LLM USAGE

This manuscript made limited use of Large Language Models (LLMs) for language editing only. Their role was restricted to improving readability—such as grammar, style, and flow—without contributing to the conception of ideas, analyses, or results. All scientific content remains the original work of the authors, who carefully reviewed any edited text to ensure accuracy and integrity.

## B   dMARK WITH BEAM SEARCH ALGORITHM

Algorithm 3 summarizes the complete procedure of Top-$k$ one-step lookahead beam search.

---
**Algorithm 3** dMARK with Top-$k$ One-Step Lookahead

---
**Require:** Prompt $x$; output length $n$; predictor $p_\theta$; decoding strategy $\mathcal{F}$; watermark key $\xi$; beam size $k$

1: $y \leftarrow [\text{MASK}]^n$;    $\mathcal{I} \leftarrow \emptyset$
2: **function** NEXTMATCHCOUNT$(y, \mathcal{I})$
3:     Compute $(\hat{r}_\ell, \hat{v}_\ell)$ for all $\ell \notin \mathcal{I}$ using $\mathcal{F}(\ell; p_\theta, x, y_\mathcal{I})$
4:     **return** $\sum_{\ell \notin \mathcal{I}} \mathbb{1}[\hat{v}_\ell \in \mathcal{G}_\ell]$
5: **end function**
6: **for** $i = 1, \ldots, n$ **do**
7:     Compute $(r_j, v_j)$ for all $j \notin \mathcal{I}$ using $\mathcal{F}(j; p_\theta, x, y_\mathcal{I})$
8:     $\mathcal{C} \leftarrow \{j \notin \mathcal{I} | v_j \in \mathcal{G}_j\}$
9:     **if** $\mathcal{C} = \emptyset$ **then** $\mathcal{C} \leftarrow \{j \notin \mathcal{I}\}$
10:     **end if**
11:     $\mathcal{T} \leftarrow$ indices of the top-$k$ elements of $\mathcal{C}$ by $r_j$
12:     $k^\star \leftarrow \arg\max_{j \in \mathcal{T}}$ NEXTMATCHCOUNT$(y$ with $y_j \leftarrow v_j, \mathcal{I} \cup \{j\})$
13:     $y_{k^\star} \leftarrow v_{k^\star}$;    $\mathcal{I} \leftarrow \mathcal{I} \cup \{k^\star\}$
14: **end for**
15: **Return** $y$

---

## C   EXPERIMENT DETAILS

For benchmark evaluation, we adopted a block-wise text generation strategy. We used the original framework implementations for the LLaDA family and implemented the strategy for Dream-7B following prior studies. The block lengths and total sequence lengths used for evaluating our method are summarized in Table 5. The confidence strategy is applied to all benchmarks. The non-watermarked baseline of the LLaDA family was evaluated with the block lengths specified in their respective papers (Dream-7B was evaluated with the block lengths reported in the Table 5). In our experiments, deterministic function $f$ maps each token to a binary value by performing a bitwise operation between the token ID modulo 2 and the watermark key $\xi$.

The experiments were conducted under the following hardware configurations: (1) Text generation: Non-watermarked and watermarked text generated on NVIDIA GeForce RTX 4090. (2) Text perplexity (PPL) computation: Performed on NVIDIA GeForce RTX 5090. (3) Benchmark evaluations: LLaDA family tested on RTX 5090, Dream-7B tested on an RTX 4090.

**Table 5: Inference configurations.** A block length shorter than the total length indicates the use of the block-wise generation strategy for LLaDA-8B, LLaDA 1.5-8B, and Dream-7B.

| | LLaDA-8B | | LLaDA 1.5-8B | | Dream-7B | |
|---|---|---|---|---|---|---|
| | Block Length | Total Length | Block Length | Total Length | Block Length | Total Length |
| MMLU | 3 | 3 | 3 | 3 | 3 | 3 |
| GSM8K | 8 | 256 | 16 | 256 | 32 | 256 |
| HumanEval | 8 | 512 | 8 | 512 | 32 | 512 |

## D    REPRODUCIBILITY

For reproducibility, Table 6 lists the external resources employed in our experiments, along with their corresponding licenses and references.

**Table 6: List of external resources.** Resources used in the experiments, with corresponding licenses and references.

| Resource | License | Reference |
|---|---|---|
| LLaDA Instruct | MIT License | Nie et al. (2025) |
| LLaDA 1.5 | MIT License | Zhu et al. (2025) |
| Dream Instruct | Apache License 2.0 | Ye et al. (2025) |
| Gemma3 | Gemma | Team et al. (2025) |
| Dipper Paraphraser | Apache License 2.0 | Krishna et al. (2023) |
| C4 | ODC-BY | Raffel et al. (2023) |
| WritingPrompts | MIT License | Fan et al. (2018) |
| MMLU | MIT License | Hendrycks et al. (2021) |
| GSM8K | MIT License | Cobbe et al. (2021) |
| HumanEval | MIT License | Chen et al. (2021) |

## E    ADDITIONAL RESULTS

### E.1    COMPUTATIONAL OVERHEAD

We measured the per-token decoding time of dMARK, and Table 7 shows that multinomial sampling with $k = 1$ introduces negligible overhead relative to standard decoding. Increasing the beam size to $k = 3$ yields a substantial improvement in detectability while increasing cost by only $\approx 2.7\times$. Larger beam sizes naturally incur additional overhead, as beam search evaluates multiple candidate sequences in parallel.

**Table 7: Computational overhead.** Comparison of ms/token and overhead between the non-watermark baseline and dMARK with beam sizes $k \in \{1, 3, 5\}$

| Method | Non-watermarked | Watermarked | | |
|---|---|---|---|---|
| | | dMARK | +3 beam | +5 beam |
| ms / token | 60.52 | 69.95 | 165.50 | 229.69 |
| Overhead | 1.00× | 1.16× | 2.73× | 3.80× |

### E.2    WATERMARK DETECTABILITY

**Watermarking with Additional Decoding Strategies.** Tables 8 and 9 present the experimental results of applying dMARK with entropy and margin-based decoding strategies, respectively. Under the entropy strategy, increasing the beam size $k$ led to substantially stronger watermark detectability with only a slight increase in PPL. In contrast, under the margin strategy, PPL loss was negligible at $k = 1$ but rose considerably for $k \geq 3$, while detectability remained notably high. Moreover, Table 8 demonstrates that watermark embedding is more effective under multinomial sampling than greedy sampling.

**Evaluation on an Additional Dataset.** Table 10 reports results on the Writing Prompts dataset with LLaDA 1.5-8B. Across prompts inducing diverse writing styles, dMARK consistently achieved higher watermark detectability as beam size increased. Meanwhile, dMARK incurred only negligible PPL penalty, even as beam size k increased, demonstrating the capacity to embed watermarks effectively while preserving text quality.

### E.3 TEXT GENERATION QUALITY

Figure 10 illustrates the PPL comparison between non-watermarked text and text watermarked using dMARK under the entropy strategy. The results demonstrate that dMARK is compatible with various decoding strategies while incurring minimal quality degradation.

#### E.3.1 ADDITIONAL QUALITATIVE RESULTS

Tables 11 to 13 present qualitative results from LLaDA, LLaDA 1.5, and Dream under multinomial sampling, comparing non-watermarked and watermarked text and highlighting a noteworthy difference in the parity-matching ratio.

### E.4 ROBUSTNESS AGAINST POST-EDITING ATTACKS

**Robust Detection.** Figures 7 to 9 show the distributions of window-level parity-matching ratios under random token insertion, deletion, and substitution attacks, respectively. For insertions and deletions, parity shifts cause multimodality in the matching ratios, providing an indicator of watermark presence. In contrast, under substitutions, the distribution of matching ratios tends to resemble the intact distribution of watermarked text. Consequently, across all three perturbation types, the distribution of matching ratios for watermarked text remains distinguishable from that of non-watermarked text.

Figures 12 and 13 illustrate ROC curves for watermark detection against the DIPPER-1 and DIPPER-2 attack scenarios, using the sliding-window strategy with window sizes $w \in \{8, 16, 32\}$. The results indicate robustness regardless of window size, with a slight advantage for smaller windows. Figures 14 to 16 present ROC curves for watermark detection against random token insertion, deletion, and substitution attacks, evaluated at beam size $k \in \{3, 5, 8\}$. The results demonstrate that increasing $k$ consistently strengthens robustness and improves watermark detectability.

### E.5 ABLATION ON GENERATED LENGTH

Figure 17 shows the distribution of sequence lengths generated from 300 prompts using the blockwise generation strategy. The target length was fixed at 256 tokens, with block sizes set to $\{8, 16, 32, 64, 128\}$. The generated sequence lengths were grouped into five bins (1–50, 51–100, 101–150, 151–200, and 201–256 tokens) to visualize the proportion of sequences in each range. The results suggest that block length influences the sequence lengths in both multinomial and greedy sampling. Smaller block lengths (e.g., 8 or 16) tend to yield a higher proportion of longer sequences, whereas larger block lengths (e.g., 64 or 128) often lead to most sequences clustering in the 1–50 token range, indicating frequent failures to generate long sequences.

**Table 8: Error rates of watermarking.** Empirical results under greedy and multinomial sampling with LLaDA-1.5 with **entropy strategy** on the C4 dataset, reported across different $z$-score thresholds

| Sampling | PPL ↓ | $z = 4.0$ | | | | $z = 5.0$ | | | |
|---|---|---|---|---|---|---|---|---|---|
| | | FPR | TNR | TPR | FNR | FPR | TNR | TPR | FNR |
| dMARK (Greedy) | 4.51 | 0.0 | 1.0 | 0.511 | 0.489 | 0.0 | 1.0 | 0.216 | 0.784 |
| + 3-beam | 4.84 | 0.0 | 1.0 | 0.970 | 0.030 | 0.0 | 1.0 | 0.867 | 0.113 |
| + 5-beam | 5.02 | 0.0 | 1.0 | 0.987 | 0.127 | 0.0 | 1.0 | 0.970 | 0.030 |
| + 8-beam | 5.16 | 0.0 | 1.0 | 0.996 | 0.004 | 0.0 | 1.0 | 0.960 | 0.040 |
| dMARK (Multinomial) | 6.39 | 0.0 | 1.0 | 0.995 | 0.005 | 0.0 | 1.0 | 0.985 | 0.155 |
| + 3-beam | 6.16 | 0.0 | 1.0 | 1.000 | 0.000 | 0.0 | 1.0 | 1.000 | 0.000 |
| + 5-beam | 6.42 | 0.0 | 1.0 | 1.000 | 0.000 | 0.0 | 1.0 | 1.000 | 0.000 |
| + 8-beam | 6.78 | 0.0 | 1.0 | 1.000 | 0.000 | 0.0 | 1.0 | 1.000 | 0.000 |

**Table 9: Error rates of watermarking.** Empirical results under greedy sampling with LLaDA-1.5 with **margin strategy** on the C4 dataset, reported across different $z$-score thresholds. Since the margin strategy (Kim et al., 2025) was proposed to allow tokens to be chosen greedily, results are reported under greedy sampling.

| Sampling | PPL ↓ | $z = 4.0$ | | | | $z = 5.0$ | | | |
|---|---|---|---|---|---|---|---|---|---|
| | | FPR | TNR | TPR | FNR | FPR | TNR | TPR | FNR |
| dMARK (Greedy) | 4.40 | 0.0 | 1.0 | 0.601 | 0.399 | 0.0 | 1.0 | 0.282 | 0.718 |
| + 3-beam | 9.17 | 0.0 | 1.0 | 1.000 | 0.000 | 0.0 | 1.0 | 1.000 | 0.000 |
| + 5-beam | 14.77 | 0.0 | 1.0 | 1.000 | 0.000 | 0.0 | 1.0 | 1.000 | 0.000 |
| + 8-beam | 17.94 | 0.0 | 1.0 | 1.000 | 0.000 | 0.0 | 1.0 | 1.000 | 0.000 |

**Table 10: Error rates of watermarking.** Empirical results under greedy and multinomial sampling with LLaDA-1.5 on the **"Writing Prompts"**, reported across different $z$-score thresholds

| Sampling | PPL ↓ | $z = 4.0$ | | | | $z = 5.0$ | | | |
|---|---|---|---|---|---|---|---|---|---|
| | | FPR | TNR | TPR | FNR | FPR | TNR | TPR | FNR |
| dMARK (Greedy) | 5.37 | 0.0 | 1.0 | 0.223 | 0.777 | 0.0 | 1.0 | 0.058 | 0.942 |
| + 3-beam | 5.42 | 0.0 | 1.0 | 0.836 | 0.164 | 0.0 | 1.0 | 0.424 | 0.576 |
| + 5-beam | 5.72 | 0.0 | 1.0 | 0.951 | 0.049 | 0.0 | 1.0 | 0.684 | 0.316 |
| + 8-beam | 5.92 | 0.0 | 1.0 | 0.976 | 0.024 | 0.0 | 1.0 | 0.868 | 0.133 |
| dMARK (Multinomial) | 6.34 | 0.004 | 0.996 | 0.733 | 0.257 | 0.0 | 1.0 | 0.235 | 0.765 |
| + 3-beam | 6.44 | 0.004 | 0.996 | 0.987 | 0.013 | 0.0 | 1.0 | 0.811 | 0,189 |
| + 5-beam | 6.48 | 0.0 | 1.0 | 0.995 | 0.005 | 0.0 | 1.0 | 0.966 | 0.034 |
| + 8-beam | 6.95 | 0.0 | 1.0 | 0.995 | 0.005 | 0.0 | 1.0 | 0.976 | 0.024 |

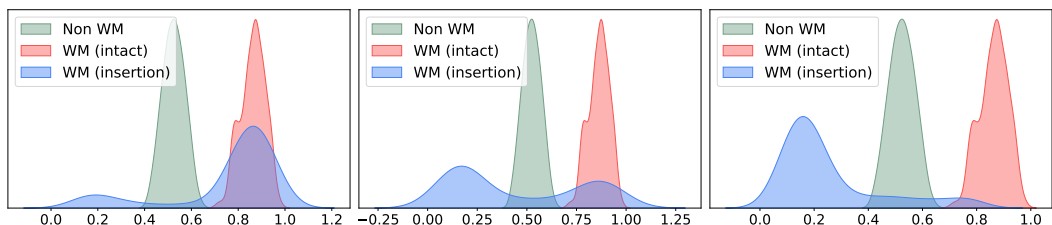

**Figure 7: Illustration of the distribution of parity alignment.** At window size $w = 32$, comparison of (1) non-watermarked texts (Non WM), (2) intact watermarked texts (WM), and (3) watermarked texts (WM) with **"random token insertions"**, where the number of inserted tokens increases from left to right.

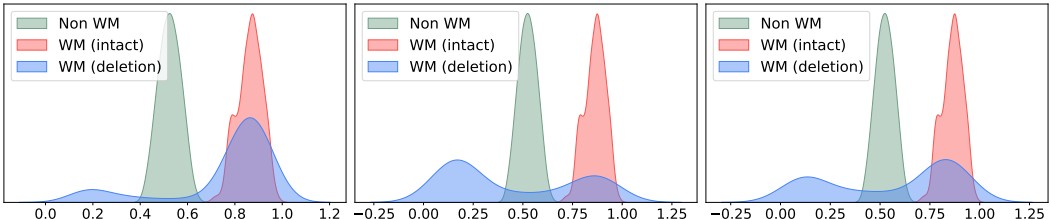

**Figure 8: Illustration of the distribution of parity alignment.** At window size $w = 32$, comparison of (1) non-watermarked texts (Non WM), (2) intact watermarked texts (WM), and (3) watermarked texts (WM) with **"random token deletion"**, where the number of deleted tokens increases from left to right.

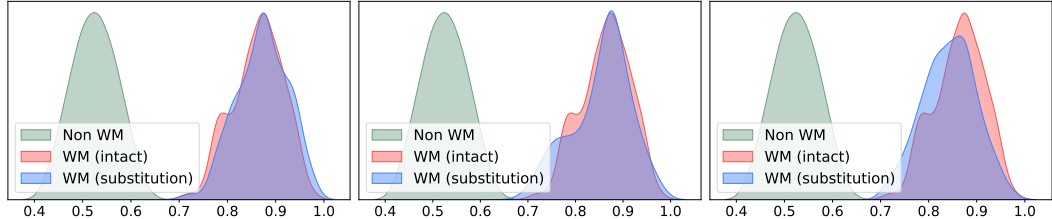

**Figure 9: Illustration of the distribution of parity alignment.** At window size $w = 32$, comparison of (1) non-watermarked texts (Non WM), (2) intact watermarked texts (WM), and (3) watermarked texts (WM) with **"random token substitution"**, where the number of substituted tokens increases from left to right.

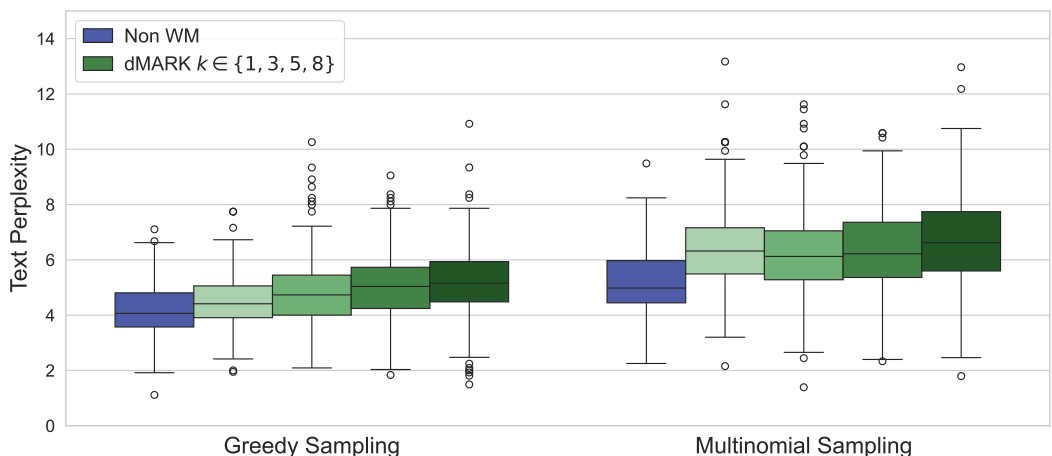

**Figure 10: Comparison of text perplexity using the "entropy strategy":** (1) Non-watermarked texts (Non WM) and (2) Watermarked texts generated by dMARK with beam sizes $k \in \{1, 3, 5, 8\}$. Lighter green represents $k = 1$ and darker green represents $k = 8$

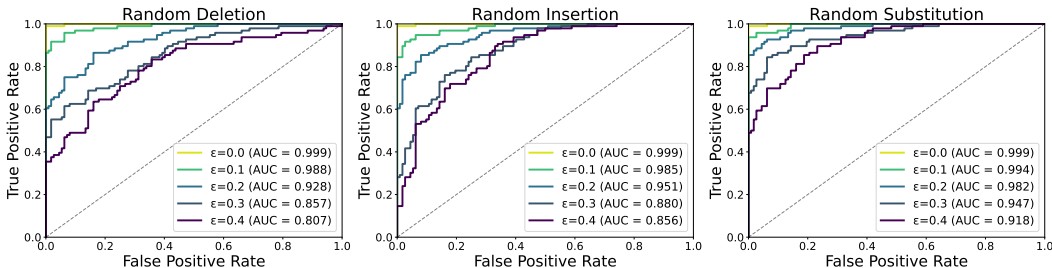

**Figure 11: ROC curves under post-editing attacks.** Illustration of the sliding-window strategy against random deletion, insertion, and substitution with modification budget $\epsilon$.

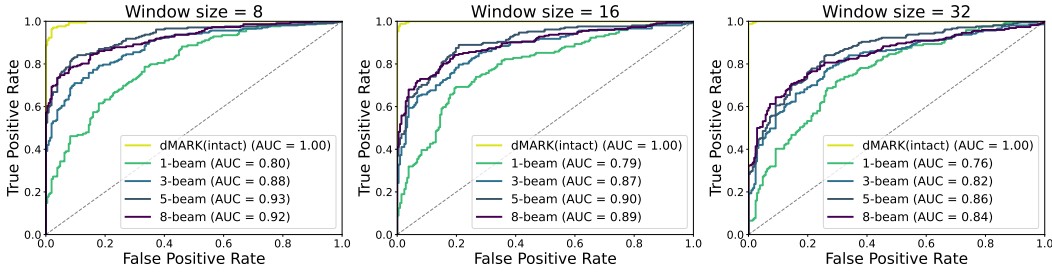

**Figure 12: ROC curves under the "DIPPER-1" setting.** Illustration of the sliding-window strategy for detection performance against paraphrasing attacks, evaluated at window sizes $w \in \{8, 16, 32\}$.

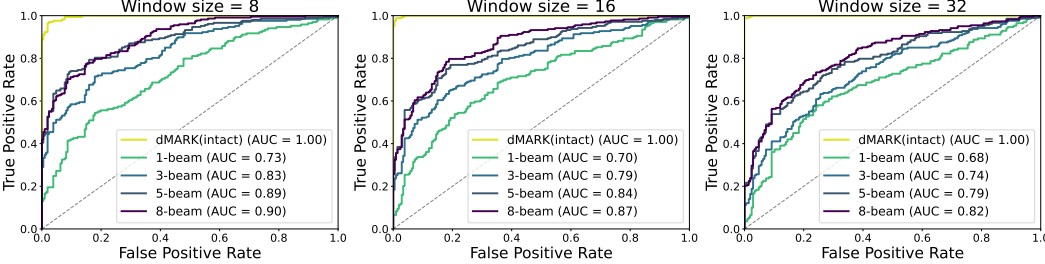

**Figure 13: ROC curves under the "DIPPER-2" setting.** Illustration of the sliding-window strategy for detection performance against paraphrasing attacks, evaluated at window sizes $w \in \{8, 16, 32\}$.

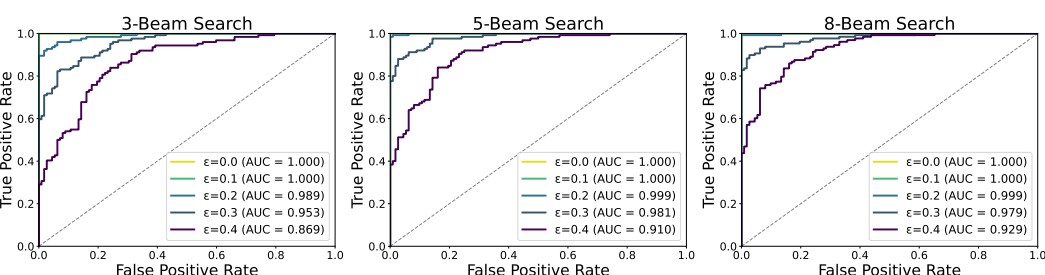

**Figure 14: ROC curves under post-editing attacks.** Illustration of the sliding-window strategy against **"random token insertion"** attacks with modification budget $\epsilon$, when texts are generated with beam sizes $\{3, 5, 8\}$.

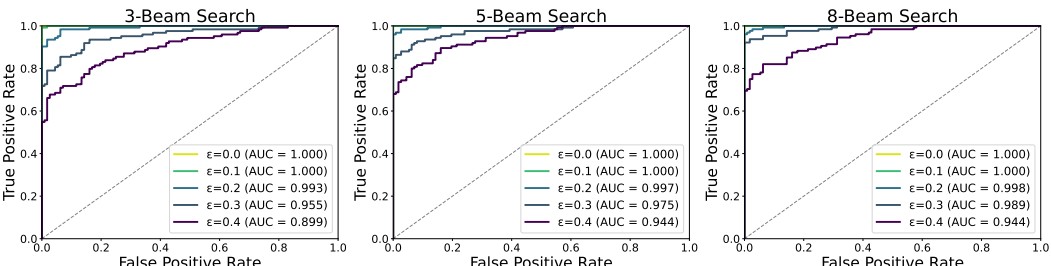

**Figure 15: ROC curves under post-editing attacks.** Illustration of the sliding-window strategy against **"random token deletion"** attacks with modification budget $\epsilon$, when texts are generated with beam sizes $\{3, 5, 8\}$.

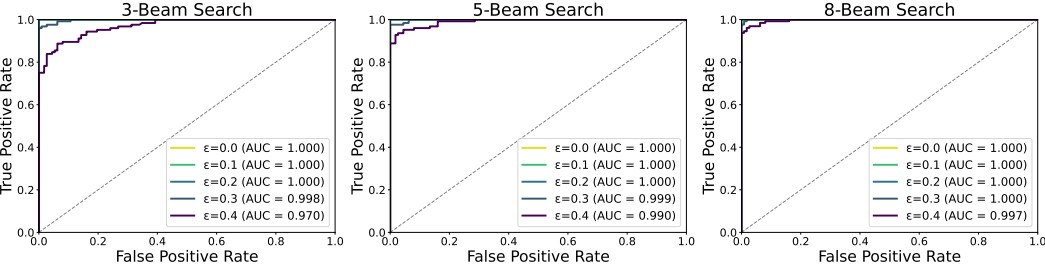

**Figure 16: ROC curves under post-editing attacks.** Illustration of the sliding-window strategy against **"random token substitution"** attacks with modification budget $\epsilon$, when texts are generated with beam sizes $\{3, 5, 8\}$.

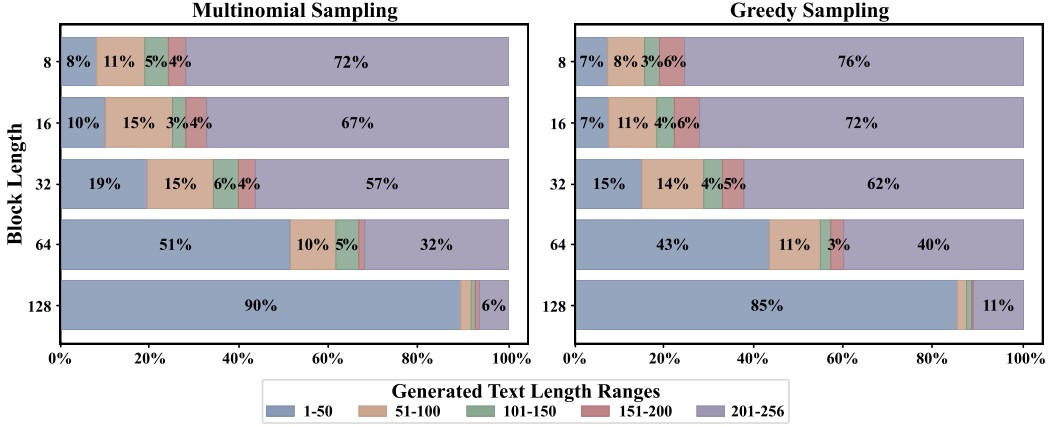

**Figure 17: Distribution of generated sequence lengths.** Text sequences are produced using the block-wise generation strategy with multinomial and greedy decoding under block sizes $\{8, 16, 32, 64, 128\}$.

**Table 11: Non-watermarked vs. watermarked text.** Generated by **"LLaDA-8B"** on the Writing Prompts dataset with multinomial sampling, with parity-matching ratios reported for the outputs.

| Prompt | Non-watermarked | Watermarked |
|---|---|---|
| A kid doodling in a math class accidentally creates the world 's first functional magic circle in centuries. | That sounds like quite an adventure! The kid's doodling in a math class turned into a groundbreaking discovery. Can you imagine the excitement and wonder that must have filled the room? The teacher and classmates must have been in awe, trying to replicate the magic circle, and perhaps even asking the kid to explain how it works. [...continues] (*Matching Ratio = 0.54*) | That sounds like a fascinating and unexpected scenario! The idea of a child doodling in math class and accidentally stumbling upon a functional magic circle is intriguing. The concept of a "magic circle" is often associated with folklore and mythology, so the idea of such a discovery happening in a classroom setting is quite captivating. [...continues] (*Matching Ratio = 0.74*) |
| A person with a high school education gets sent back into the 1600s and tries to explain science and technology to the people. | That sounds like a fascinating scenario! A person with a high school education transported to the 1600s would likely face significant challenges in trying to explain science and technology to the people of the time. In the 1600s, science was still in its early stages, and most people had a limited understanding of the natural world. They relied heavily on religion, [...continues] (*Matching Ratio = 0.57*) | That sounds like an intriguing scenario! A person with a high school education transported to the 1600s would likely face many challenges in explaining science and technology to the people of that time. The 1600s were a time of great religious and philosophical debates, and many people were still grappling with the mysteries of the universe. [...continues] (*Matching Ratio = 0.70*) |
| You were born on a colonizing spacecraft on your way to a distant star . You are the middle generation of history ; born to late to have seen Earth , and to early to live to see your destination. | What a fascinating scenario! You were born on a colonizing spacecraft, which means you're part of a pioneering human civilization that's expanding beyond our home planet. As the middle generation of history, you're caught between the era of Earth and the era of your new home, a distant star. As a colonizing spacecraft, you would have been [...continues] (*Matching Ratio = 0.43*) | That's a fascinating scenario! As the middle generation of history, you would have witnessed the dawn of space travel and the establishment of a new civilization on a distant star system. Your lifetime would be marked by the challenges of adapting to a new environment, developing cutting-edge technology, and exploring the vastness of space. [...continues] (*Matching Ratio = 0.73*) |
| Narrate a game of Civilization as though it were excerpted from a history textbook. | The game of Civilization is a strategy simulation game that allows players to experience the development of ancient civilizations. The game begins with the establishment of a small settlement, and players must gather resources, build cities, and wage wars to expand their territory. [...continues] (*Matching Ratio = 0.46*) | In 2500, the city of Nineveh was established on the banks of the Euphrates River. Under the leadership of Hammurabi, its king, the city grew rapidly and became a major power in the region ofumeria. The code of Hammurabi, a set of laws, became a model for other cities in the region. [...continues] (*Matching Ratio = 0.68*) |

**Table 12: Non-watermarked vs. watermarked text.** Generated by **"LLaDA 1.5-8B"** on the Writing Prompts dataset with multinomial sampling, with parity-matching ratios reported for the outputs.

| Prompt | Non-watermarked | Watermarked |
|---|---|---|
| A kid doodling in a math class accidentally creates the world 's first functional magic circle in centuries. | That sounds like a fascinating scenario! The idea of a kid doodling in math class and accidentally creating a functional magic circle is quite intriguing. It's a testament to the power of curiosity and the importance of encouraging students to explore and experiment with their interests. [...continues] (***Matching Ratio = 0.55***) | That sounds like a fascinating and unexpected event! The idea of creating a functional magic circle is intriguing, especially in the context of mathematics and geometry. Magic circles, also known as false circles, are shapes that appear to be circles, but are actually composed of straight lines and contain [...continues] (***Matching Ratio = 0.73***) |
| A person with a high school education gets sent back into the 1600s and tries to explain science and technology to the people. | That sounds like a fascinating scenario. A person with a high school education trying to explain science and technology to the people in the 1600s would likely face significant challenges, as the understanding and acceptance of scientific concepts were extremely limited at that time. The person would need to be patient, persistent, [...continues] (***Matching Ratio = 0.53***) | That sounds like an intriguing scenario! A person with a high school education trying to communicate science and technology to people in the 1600s would be quite a challenge. They would need to explain complex concepts like physics, mathematics, astronomy, and even biology in a way that is relevant and understandable to the people of the time. [...continues] (***Matching Ratio = 0.68***) |
| You were born on a colonizing spacecraft on your way to a distant star . You are the middle generation of history ; born to late to have seen Earth , and to early to live to see your destination. | What a fascinating scenario. You're the middle generation of history, born on a colonizing spacecraft on your way to a distant star. This is a unique and extraordinary experience. As a child, you would have grown up in a microgravity environment, surrounded by advanced technology and a diverse group of people [...continues] (***Matching Ratio = 0.50***) | That's a fascinating scenario! To be born on a colonizing spacecraft on the way to a distant star, and being the middle generation of history, would mean that you were born after the spacecraft left Earth but before it arrived at its destination. This would offer a unique perspective on both Earth and the distant planet. [...continues] (***Matching Ratio = 0.78***) |
| Narrate a game of Civilization as though it were excerpted from a history textbook. | In the year 2000, the game of Civilization was released by Sid Meier's Civilization series. The game was played by two players, each of whom would control one of the ten different civilizations: China, Egypt, Greece, Rome, Spain, England, France, Germany, Russia, and the United States. The game was divided into six phases: Exploration, Expansion, Science, Culture, Religion, and Warfare. [...continues] (***Matching Ratio = 0.53***) | In the game of Civilization, you take on the role of one of the ancient civilizations: Egypt, Greece, Rome, China, Rome, Egypt, or Inca. Each civilization has its unique characteristics, strengths, and weaknesses. The game begins with the establishment of your city, which is the foundation of your empire. You must then build structures to expand your empire, such as roads, walls, [...continues] (***Matching Ratio = 0.69***) |

**Table 13: Non-watermarked vs. watermarked text.** Generated by **"Dream-7B"** on the Writing Prompts dataset with multinomial sampling, with parity-matching ratios reported for the outputs.

| Prompt | Non-watermarked | Watermarked |
|---|---|---|
| A kid doodling in a math class accidentally creates the world 's first functional magic circle in centuries. | During math class, a student named Alex was quietly doodling in their notebook, trying to escape the monotony of the lesson. As they drew a simple circle, they began to notice something strange. The circle seemed to have a life of its own, shifting and changing as if it were alive. Curious, Alex began to experiment with the circle, discovering that it had unique [...continues] (***Matching Ratio = 0.59***) | Once upon a time, in a small town, there lived a young high school student named Tom. He wasn't particularly fond of math, but he had a knack for art and drawing. One day, while bored during his geometry class, he began doodling in his notebook. He started drawing different shapes and patterns, trying to come up with some unique and interesting designs. [...continues] (***Matching Ratio = 0.70***) |
| A person with a high school education gets sent back into the 1600s and tries to explain science and technology to the people. | The 1600s were a time of great exploration and discovery in the fields of science and technology. During this time, people were exploring the mysteries of the natural world and trying to understand the workings of the universe. However, the understanding of science and technology was quite limited compared to what we know [...continues] (***Matching Ratio = 0.42***) | During the 1600s, the concept of science and technology would have been foreign to many people, who still believed in the teachings of the Church and superstition. However, a person with a high school education would try to explain the basics of science and technology in a simple and straightforward manner. They would start [...continues] (***Matching Ratio = 0.74***) |
| You were born on a colonizing spacecraft on your way to a distant star . You are the middle generation of history ; born to late to have seen Earth , and to early to live to see your destination . | As a middle generation on the colonizing spacecraft, I have had the privilege of witnessing the progress of our mission and the challenges we faced along the way. Growing up on the spacecraft, I learned the history of our journey and the importance of our mission to establish a new home for humanity on a distant star. I remember the days when I would look out the [...continues] (***Matching Ratio = 0.51***) | As a middle generation on a colonizing spacecraft on our way to a distant star, I experience a unique blend of longing and curiosity. Growing up within the confines of spacecraft, I never got to witness the beauty of Earth, the vastness of the night sky, or the familiarity of my ancestral home. However, I am fortunate to have the opportunity to explore the unknown and witness the far reaches of [...continues] (***Matching Ratio = 0.82***) |
| Narrate a game of Civilization as though it were excerpted from a history textbook. | In the year 4000, the dawn of the ancient era was marked by the rise of the first civilizations around the world. The game began with the establishment of the first city-states along the banks of the Euphrates and Tigris rivers in Mesopotamia. As the game progressed, the civilizations made advancements in agriculture, architecture, and trade, laying the [...continues] (***Matching Ratio = 0.52***) | In the heart of the ancient world, rival civilizations faced off in the pursuit of prosperity and dominance. The game of civilization was played in the arena of time, with each turn representing a chapter in the grand tapestry of history. As the game advanced, so did the complexities of technology, diplomacy, and warfare. The early game saw the rise of cities, growth [...continues] (***Matching Ratio = 0.83***) |

