# OpenReview forum: "dMARK: Decoding-Guided Watermarking for Discrete Diffusion Language Models"
_ICLR.cc/2026/Conference — Submitted to ICLR 2026_

### Official Review · Reviewer_JtBe · 2025-10-15

**Soundness:** 3
**Presentation:** 1
**Contribution:** 2
**Rating:** 2
**Confidence:** 5

**Summary:**

This paper proposes a watermarking method for discrete diffusion language models, where the decoding order is controlled based on binary hash rules to embed watermark signals. This method is compatible with common decoding strategies and can be further enhanced by beam search. Experimental results across multiple models and datasets demonstrate its effectiveness.

**Strengths:**

1. The paper explores watermarking schemes for the decoding process of dLLMs.

2. The paper provides experimental analysis from the perspectives of detectability, text quality, and robustness.

**Weaknesses:**

1. The writing is quite rough; for example, references to appendices in the main text lack chapter numbers, and the wording is imprecise. The abbreviation for discrete diffusion language models should be "ddlms," not "dllms," which refers to "diffusion large language models."
2. The logic in the Introduction section is somewhat disorganized, alternating between discussions of LLMs and dLLMs (e.g., the first paragraph mentions dLLMs, the second paragraph shifts to LLMs, the third paragraph discusses related work on LLMs, and only towards the end does it address dLLMs). The motivation for the paper is not clearly presented.
3. The description of the proposed method is rather simplistic, with much of it being based on prior work, such as dLLM decoding, beam search sampling, and window-based detection, making it hard to discern the core innovation.
4. There is a lack of an Ethics Statement and Reproducibility Statement sections.

**Questions:**

1. What are the fundamental differences between this approach and KGW? It seems to be merely a substitution of the hash scheme.
2. The input to the hash function only consists of the watermark key and tokens from the vocabulary. Does this mean the segmentation across all positions is identical?
3. Does the "beam search" mentioned in the paper refer to sampling tokens from multiple positions at the same timestep?

---

> ### Author Response · Authors · 2025-11-19
> **Rebuttal**
>
> We appreciate your valuable comments. We address your feedback and questions below.
>
> ---
> > **[W1]** Issues regarding writing style and abbreviation usage
>
> Thank you for pointing this out. We have revised the manuscript to improve clarity, including adding chapter numbers to all appendix references and fixing imprecise wording. Regarding the abbreviation, we acknowledge the potential confusion. While the literal abbreviation of discrete diffusion language models would be “ddLMs,” we follow the convention used in the prior survey [1] and adopt “dLLMs” throughout the paper. In the revised manuscript, we define this abbreviation at its first appearance and use it consistently.
>
> [1] Discrete Diffusion in Large Language and Multimodal Models: A Survey, Arxiv, 2025
>
> > **[W2]** Issues with the organization of the Introduction and the clarity of the paper’s motivation
>
> Thank you for this helpful observation. We agree that the earlier version of the Introduction mixed discussions of LLMs and dLLMs, which made the motivation less clear. In the revised manuscript, we have reorganized the **Section 1** by separating the discussion of LLMs and dLLMs to provide a clearer motivation for our work. We believe this restructuring clarifies the motivation and improves the logical flow of the Introduction.
>
> > **[W3], [Q1]** Concern about dependence on prior components and the need to clarify our fundamental difference from KGW.
>
> Thank you for raising this point. While our method builds on standard components such as dLLM decoding and beam search, dMARK is fundamentally different in that it embeds watermarks without altering any token probabilities, exploiting the order-agnostic structure of dLLMs. Existing watermarking methods modify or reweight token probabilities, whereas dMARK achieves detectability purely through decoding-order guidance, which has not been explored in prior work.
>
> Our beam-search variant also differs from conventional beam search: instead of expanding full sequence continuations, we use a one-step lookahead over positions and select the candidate that maximizes future parity matches. This decoding-guided criterion is unique to dLLMs and to our watermarking formulation.
>
> Similarly, our sliding-window detection differs from prior window-based methods that locate watermarked segments or recover seeds [1-4]. We instead measure local parity-matching ratios and use their distribution for detection-an approach specific to our decoding-guided embedding mechanism.
>
> [1] WaterSeeker: Pioneering Efficient Detection of Watermarked Segments in Large Documents, ACL, 2025
> [2] BiMark: Unbiased Multilayer Watermarking for Large Language Models, ICML, 2025
> [3] On the Reliability of Watermarks for Large Language Models, ICLR, 2024
> [4] Scalable watermarking for identifying large language model outputs, Nature, 2024
>
>
> > **[W4]** Lack of Ethics and Reproducibility Statements
>
> We appreciate the feedback. We have updated the manuscript by adding Ethics Statement and Reproducibility Statement sections.
>
> > **[Q2]** Hash function’s input implying identical segmentation across positions
>
> Thank you for the question. It is correct that the hash function induces a fixed, position-independent partition of the vocabulary. However, this does not imply identical matching sets across positions. A token belongs to the matching set at position $i$ only if its hash value also matches the parity of the position index. Thus, while the segmentation of the vocabulary is fixed, the effective matching set varies with $i$.
>
> Importantly, this design also addresses a new challenge specific to dLLMs: because decoding is not left-to-right, the previous token (typically hashed in ARM-based watermarking) is often undefined or unavailable. Our approach resolves this by using position-agnostic hashing (hashing token values only) combined with position-dependent parity alignment. This makes watermarking feasible in an order-agnostic decoding setting.
>
> Finally, sliding-window detection further provides robustness to shift-based perturbations: insertions or deletions shift position indices, altering parity alignment patterns. By monitoring matching ratios across windows, our detector still reliably identifies the watermark.
>
>
> > **[Q3]** Clarify the beam search definition
>
> Thank you for the question. Our use of the term “beam search’’ does not refer to sampling tokens from multiple positions at the same timestep. Instead, it is a lookahead-based variant of dMARK: at each decoding step, we identify the top-$k$ candidate positions (and their predicted tokens) that satisfy the parity alignment condition. For each candidate, we then perform a one-step lookahead by querying the model on the remaining masked positions and counting how many predicted tokens fall into the parity-matching set. We select the candidate that maximizes this quantity.
>
> Thus, our “beam search’’ operates over future parity alignment, not over multiple simultaneous token generations.

---

> > ### Comment · Reviewer_JtBe · 2025-11-26
> >
> > Thank you for the authors' clarification. Although the authors have made extensive revisions based on the original manuscript, this also indicates that there is still considerable room for improvement. In addition, I still have reservations about the novelty of the paper. Therefore, I will maintain my original score.

---

### Official Review · Reviewer_DGd2 · 2025-10-31

**Soundness:** 3
**Presentation:** 3
**Contribution:** 3
**Rating:** 4
**Confidence:** 4

**Summary:**

The authors introduced a watermarking method for discrete diffusion language models (dLLMS) called dMARK. The authors claimed that the proposed method is the first decoding-guided watermarking method for such models. dMARK embeds watermark signals without altering token probabilities, ensuring strong detectability and minimal quality degradation. Experiments were done demonstrating dMARK method’s performance and robustness.

**Strengths:**

1. The proposed method is designed to be effectively combined with any decoding strategy, the ones listed in the paper include confidence, entropy, margin, and greedy-based rules.

2. The authors claim that this is the first watermarking method done on dLLMs, signaling research significance and a current need for such a method.

3. The overall motivation of this paper is solid - identifying a gap in research on a fast-rising new variant of LLMs (dLLMs) and exploiting new vulnerabilities introduced by the unique design of such models: the order-agnostic property forces decoding strategy to be an essential design choice.

**Weaknesses:**

1. The paragraph on robustness to post-editing is very short, it would be valuable to see more discussions on robustness evaluations against more sophisticated post-editing transformations instead of just insert/delete/substitution.

2. Although it is claimed to be the first watermarking method done on dLLMs, comparing against other watermarking methods on other language models can help demonstrate the effectiveness of the method.

3. The proposed method can increase detectability but also increase inference cost and PPL. More analysis on the computational overhead is needed.

4. It would be helpful to have a method figure, describing a pipeline framework of the proposed method from input to output. The only figure the authors demonstrated before the experiments was an output sample watermarked paragraph.

**Questions:**

1. According to Table 3, the text generation quality between the dMARK method and the dMARK+ 3-beam does not seem to vary much on MMLU and GSM-8K. Is there any extended discussion on why that is?

2. What is the performance on short answers (small number of output tokens)?

---

> ### Author Response · Authors · 2025-11-19
> **Rebuttal (1/2)**
>
> We appreciate your constructive comments. We address your feedback and questions below.
>
> ---
> > **[W1]** The need to address robustness under more sophisticated post-editing transformations
>
> Thank you for the suggestion. Beyond simple insert/delete/substitute operations, we also evaluate robustness under paraphrasing attacks, which constitute a significantly more complex form of post-editing. To this end, we use DIPPER, an 11B-parameter paraphrase-generation model designed specifically to stress-test watermark detection systems. We consider two configurations of DIPPER in our paraphrasing evaluation. DIPPER-1 performs paraphrasing at predefined ratios via lexical modification, whereas DIPPER-2 applies ratio-adjusted lexical modification together with an additional 10% order diversity.
> As shown in Figure 5 (Section 4) and the additional table provided in the supplement, dMARK maintains high AUC across a wide range of paraphrasing intensities, demonstrating robustness against substantially more sophisticated transformations than token-level edits. In addition, we have expanded our evaluation of post-editing robustness in the revision. Specifically, we compare dMARK with existing watermarking schemes under insertion, deletion, and substitution attacks, with the results included in **Figure 4 (Section 4)**.
>
> - Dipper-1
>
> |                        | **AUC ↑** |       |       |       |       |
> |------------------------|---------|-------|-------|-------|-------|
> | **Modification Ratio** | **0%**      | **10%**   | **20%**   | **30%**   | **40%**   |
> | KGW                    | 0.960   | 0.740 | 0.785 | 0.715 | 0.685 |
> | PATTERN-MARK           | 0.988   | 0.879 | 0.871 | 0.791 | 0.755 |
> | dMARK + 3beam          | **0.999**   | **0.884** | **0.907** | **0.822** | **0.757** |
>
> - Dipper-2 (additional 10% order diversity)
>
> |                        | **AUC ↑** |       |       |       |       |
> |------------------------|---------|-------|-------|-------|-------|
> | **Modification Ratio** | **0%**      | **10%**   | **20%**   | **30%**   | **40%**   |
> | KGW                    | 0.960   | 0.693 | 0.735 | 0.660 | 0.679 |
> | PATTERN-MARK           | 0.988   | 0.822 | 0.843 | **0.810** | **0.762** |
> | dMARK + 3beam          | **0.999**   | **0.835** | **0.879** | 0.786 | 0.757 |
>
> We have further extended our evaluation by incorporating a comparison against existing watermarking methods under DIPPER-based paraphrasing attacks, updating **Figure 5 (Section 4)** to reflect these results.
>
> > **[W2]** The need to compare with existing watermarking methods
>
> Thank you for the suggestion. We agree that comparing dMARK with existing watermarking approaches is important for demonstrating its effectiveness. To address this, we conducted additional experiments comparing dMARK with (1) KGW, an ARM-style watermarking method applied to dLLMs by enforcing left-to-right generation, and (2) PATTERN-MARK, the watermarking method designed for order-agnostic models.
>
> | **Greedy**        | **PPL ↓**  | **TPR@FPR ↑** |        |          |           | **Multinomial**   | **PPL ↓**  | **TPR@FPR ↑** |        |          |           |
> |-------------------|----------|:-----------:|:------:|:--------:|:---------:|-------------------|----------|:-----------:|:------:|:--------:|:---------:|
> |                   |          | **10%**     | **1%** | **0.1%** | **0.01%** |                   |          | **10%**     | **1%** | **0.1%** | **0.01%** |
> | Non-Watermark | 4.03 |             |        |          |           | Non-Watermark | 4.21 |             |        |          |           |
> | KGW               | 5.83     | **100.00**      | **100.00** | 98.52    | **97.78**     | KGW               | 7.87     | **100.00**      | 99.21  | 99.21    | 98.41     |
> | PATTERN-MARK      | 5.86     | 96.26       | 91.59  | 87.38    | 78.97     | PATTERN-MARK      | 7.69     | 98.99       | 95.96  | 91.41    | 83.33     |
> | dMARK             | **4.44**     | 97.86       | 91.98  | 76.47    | 60.96     | dMARK             | **5.27**     | **100.00**      | **100.00** | 99.41    | 95.29     |
> | dMARK + 3beam     | 4.75     | **100.00**      | 99.54  | **98.62**    | 97.25     | dMARK + 3beam     | 5.40     | **100.00**      | **100.00** | **100.00**   | **100.00**    |
>
> These results have been added to **Table 2 in Section 4**. Overall, the comparisons show that dMARK achieves stronger detectability with noticeably less quality degradation than both alternatives.

---

> > ### Author Response · Authors · 2025-11-19
> > **Rebuttal (2/2)**
> >
> > > **[W3]** The need for additional analysis of computational overhead introduced by the proposed method
> >
> > Thank you for the suggestion. We agree that analyzing computational overhead is important. We measured the per-token decoding time of dMARK for $k\in\\{1,3,5\\}$, and the results show that multinomial sampling with $k{=}1$ adds only modest overhead.
> > As shown in Table 2 of the manuscript (Section 4), dMARK achieves sufficiently strong detectability even when using multinomial sampling with $k{=}1$. Importantly, $k{=}3$ provides a dramatic improvement in detectability (regardless of the sampling) while incurring only moderate additional cost ($\approx 2.7 \times$). Higher beam sizes naturally incur more overhead, as beam search evaluates multiple candidate positions in parallel, but this additional cost yields correspondingly higher detectability.
> >
> > | **Method**   | **Non-watermark** |   **dMARK**  |  **+3 beam**  |  **+5 beam**  |
> > |----------|:-------------:|:--------:|:---------:|:---------:|
> > | ms/token | 60.52 ms      | 69.95 ms | 165.50 ms | 229.69 ms |
> > | Overhead | 1.00x         | 1.16x    | 2.73x     | 3.80x     |
> >
> > We have added the relevant explanation and the table to the **Appendix E.1**.
> >
> >
> > > **[W4]** It would be helpful to include an overview figure to clarify the proposed method.
> >
> > Thank you for the suggestion. We agree that an overview figure would greatly improve clarity. In the revised manuscript, we have added a new **figure 1 in Section 1** that provides a concise comparison of standard autoregressive watermarking, conventional dLLM decoding, and the decoding order used in dMARK. Although not a full pipeline diagram, this comparative illustration highlights the key differences among the three decoding procedures and we believe it meaningfully clarifies the core idea of our method.
> >
> > > **[Q1]** Discussion on the minimal performance difference between dMARK and dMARK+3-beam on MMLU and GSM-8K
> >
> > Thank you for the question. The small performance gap between dMARK and dMARK+3-beam on MMLU and GSM8K stems from a key property of our method: dMARK never modifies the underlying probability distribution, so the probability ranking of answer-critical tokens is fully preserved.
> >
> > For MMLU, answers are short and typically depend on a small number of decisive tokens; since their probabilities are unchanged, both dMARK and dMARK+3-beam select the same answers in most cases. GSM8K generates longer solutions, but the decisive reasoning tokens are likewise preserved because the probability structure is untouched. As a result, both tasks show minimal performance differences despite the stronger watermark alignment achieved by beam search.
> >
> >
> > > **[Q2]** Performance on short answer outputs
> >
> > Thank you for the question. We have already included an analysis of watermark performance as a function of answer length (**Figure 6 in Section 5**). As shown there, detectability is limited when the number of generated tokens is below 50, but it improves rapidly as the sequence length increases. This behavior is consistent with prior work [1–3] and reflects a general property of statistical watermarking: longer outputs provide more evidence for detection. Importantly, our method does not require any modification to support short answers; the same detector applies, and detection strength scales naturally with output length.
> >
> > [1] GaussMark: A Practical Approach for Structural Watermarking of Language Models, ICML, 2025
> > [2] Provable Robust Watermarking for AI-Generated Text, ICLR, 2024
> > [3] Scalable watermarking for identifying large language model outputs, Nature, 2024

---

### Official Review · Reviewer_mPfw · 2025-11-01

**Soundness:** 2
**Presentation:** 2
**Contribution:** 2
**Rating:** 4
**Confidence:** 3

**Summary:**

This paper proposes a watermarking method for dLLMs. For dLLMs, the generation process is no longer auto-regressive but in a specific order decided by a reward function per token position. The authors modify the generation order by prioritizing those positions of which the to-be-generated token's hash value (determined by a watermarking key) is 1 (mod 2). Detection is performed with a parity‑matching ratio and one‑sided $z$‑test. The authors also propose a beam search variant. Experiments are conducted on LLaDA‑8B, LLaDA‑1.5‑8B, and Dream‑7B, showing low FPR with competitive text quality. Robustness under random token edits and paraphrasing is also tested and discussed.

**Strengths:**

1. The paper proposes the first method that deals with watermarking dLLMs. The order of generation is modified, which makes sense in the scenario of dLLMs where the order matters very much.

2. The self-hashing algorithm is easy to implement. Also, it is robust to edit.

**Weaknesses:**

1. The idea of self-hashing has already been conducted long ago (for example, *A Watermark for Large Language Models* Algorithm 3). This self-hashing algorithm in the work is very similar to theirs. The algorithm works like rejecting all other $n - 1$ tokens and accepting only $1$ token per generation, and the rejection probability is higher if the sampled token is in $\mathcal{R}$ (which is what *A Watermark for Large Language Models* Algorithm 3 is doing). Although the conduction may be different, the final outcome is very similar. Also, the detection method is exactly the same. In that sense, the novelty of this algorithm is undermined.

2. Theory: watermarking algorithm's "unbiasedness". There are two concerns. One major concern is that the algorithm's "unbiasedness" is not proved. I know the proof might seem trivial to the authors, but that claim must be proved in a self-contained manner for completeness. One minor concern is that the idea of "unbiasedness" is only a marginal unbiasedness with respect to key $\xi$ (for example, see *Towards Better Statistical Understanding of Watermarking LLMs*). More specifically, only $\mathbb{E}_{\xi}[p_{\text{watermarked}}(y|x, \xi)] = p_{\text{unwatermarked}}(y|x)$ holds rather than $p_{\text{watermarked}}(y|x, \xi) = p_{\text{unwatermarked}}(y|x)$. I think the authors should be more precise. Adjusting the priority of generation is actually equivalent to filtering out those tokens in $\mathcal{R}$, and that should be clearly addressed.

3. The text quality seems to degrade by a large degree. I think that might be due to modifying the order of generation, which is a core ingredient in developing high-quality dLLMs.

**Questions:**

1. Please add more dicsussions on the "unbiasedness" of this algorithm. A proof is necessary.

2. Please address the difference between your algorithm and the self-hashing algorithm of Kirchenbauer et al. 2023.

---

> ### Author Response · Authors · 2025-11-19
> **Rebuttal (1/2)**
>
> We appreciate your insightful comments. We address your feedback and questions below.
>
> ---
> > **[W1], [Q2]** Concerns that the proposed method closely resembles prior self-hashing watermarking [1], undermining novelty, and a request to clarify the algorithmic differences
>
> Thank you for raising this point. We agree that our method is conceptually related to prior hashing-based watermarking (e.g., Kirchenbauer et al., 2023), as most watermarking approaches partition the vocabulary into two groups and analyze their frequency. We now clarify this connection in the revised paper, **Section 3.3 (lines 201-206)**.
> However, our method differs in several essential ways:
> - **Model setting**: Prior self-hashing methods rely on the autoregressive next-token context (hashing the previous token or n-gram). In contrast, dMARK targets discrete diffusion LLMs, which generate tokens in arbitrary orders. Applying watermarking in this setting is non-trivial because there is no fixed previous-token context to hash.
> - **No probability biasing or rejection**: The method of [1] increases the probability of “green’’ tokens (or rejects others). dMARK does not modify or bias token probabilities at any step. The model distribution is preserved exactly.
> - **Decoding-based embedding**: Instead of probability adjustments, dMARK embeds the watermark by guiding the unmasking order, a degree of freedom unique to dLLMs. This mechanism does not exist in autoregressive models and is fundamentally different from self-hashing.
> - **Detection rule**: Our parity-based detection does not depend on previous tokens or n-grams, unlike the context-dependent hashing in [1].
>
> Recent studies [2–3] have also attempted to adapt existing Autoregressive (AR) hashing-based watermarking methods to dLLMs. However, due to the structural characteristics of dLLMs, such AR approaches are difficult to apply directly, and even when they can be applied, they remain fundamentally the same in that they embed signals by adjusting token probabilities. Thus, although motivated by prior hashing ideas, our method introduces a new decoding-guided watermarking framework tailored specifically to dLLMs.
>
> [1] A watermark for large language models. ICML, 2023
> [2] DMark: Order-Agnostic Watermarking for Diffusion Large Language Models, Arxiv, 2025
> [3] Watermarking Diffusion Language Models, Arxiv, 2025
>
> > **[W2] , [Q1]** The need for a proof of the algorithm’s unbiasedness
>
> Thank you for pointing this out. In our setting, unbiasedness means that dMARK does not modify or reweight any token probabilities, unlike prior hashing-based methods (e.g., Kirchenbauer et al., 2023), which boost or suppress tokens and therefore only satisfy marginal unbiasedness.
>
> In dMARK, watermarking is introduced solely through the decoding order, not probability adjustment. For every step, we use the model’s conditional probabilities exactly as-is. No tokens are filtered or rejected, $\mathcal{G}$ tokens are simply decoded earlier.
> We added a short justification showing that, due to the order-agnostic factorization of dLLMs, changing the unmasking order alone leaves the model’s induced distribution unchanged.
>
> We assume an ideal predictor $p_\theta$ that exactly matches the true conditional distributions for any visible index set $\mathcal{I}$ , i.e.,
>
> $
> p_\theta(y_i \mid y_{\mathcal{I}}, x) = p_{\text{data}}(y_i \mid y_{\mathcal{I}}, x) \quad \forall i, \mathcal{I}.
> $
>
> Consider any permutation $\pi$ of $\\{1, \dots, n\\}$, and the following sampling procedure: at step $t$, we sample $y_{\pi(t)}$ from
>
> $
> p_\theta\bigl(\cdot \mid y_{\pi(1)},\dots,y_{\pi(t-1)}, x\bigr).
> $
>
> The probability that this procedure generates a particular sequence $y$ is
>
> $
> p_{\theta}(y|x) =\prod_{t=1}^np_\theta\bigl(y_{\pi(t)} \mid y_{\pi(<t)}, x\bigr)=\prod_{t=1}^n p_{\text{data}}\bigl(y_{\pi(t)} \mid y_{\pi(<t)}, x\bigr)= p_{\text{data}}(y \mid x),$
>
> where the last equality follows from the chain rule applied to $p_{\text{data}}$ along the permutation $\pi$. Since this holds for any permutation $\pi$, the resulting sampling distribution is exactly $p_{\text{data}}(\cdot \mid x)$ and is independent of the unmasking order.

---

> > ### Author Response · Authors · 2025-11-19
> > **Rebuttal (2/2)**
> >
> > > **[W3]** Concern about noticeable text quality degradation, potentially caused by altering the generation order in dLLMs
> >
> > We agree that watermarking introduces an inherent detectability-quality trade-off, a well-known limitation in prior work. In our experiments, the quality impact of dMARK is moderate and mainly observed in low-entropy tasks such as code generation, while degradation on MMLU and GSM8K remains small.
> >
> > To contextualize this trade-off, we compare dMARK against PATTERN-MARK (the watermarking method for order-agnostic models), as well as KGW (an ARM-style watermarking adapted to dLLMs by enforcing a left-to-right decoding order). Although the latter is not a fully fair comparison, it is the only practical way to apply ARM watermarking to dLLMs. In both cases, dMARK incurs noticeably less quality degradation because it does not modify token probabilities, whereas PATTERN-MARK and ARM-style methods rely on probability biasing. The remaining quality loss in dMARK arises solely from adjusting the decoding order and reflects the inherent quality-detectability trade-off rather than distributional distortion, and represents the unavoidable quality-detection trade-off.
> >
> >
> > - LLaDA
> >
> > |               | **Greedy** |           |               | **Multinomial** |           |               |
> > |---------------|:----------:|:---------:|:-------------:|:---------------:|:---------:|:-------------:|
> > | **Method**    | **MMLU ↑**   | **GSM8K ↑** | **HumanEval ↑** | **MMLU ↑**        | **GSM8K ↑** | **HumanEval ↑** |
> > | Non-watermark | 0.648      | 0.797     | 0.427         | 0.594           | 0.775     | 0.360         |
> > | KGW           | 0.558      | 0.662     | 0.092         | 0.520           | 0.464     | 0.055         |
> > | PATTERN-MARK  | 0.570      | 0.635     | 0.134         | 0.532           | 0.438     | 0.073         |
> > | dMARK         | **0.647**      | **0.787**     | **0.280**         | **0.588**           | **0.735**     | **0.226**         |
> > | dMARK + 3beam | **0.647**      | 0.771     | 0.268         | 0.580           | 0.678     | 0.152         |
> >
> > - LLaDA 1.5
> >
> > |               | **Greedy** |           |               | **Multinomial** |           |               |
> > |---------------|:----------:|:---------:|:-------------:|:---------------:|:---------:|:-------------:|
> > | **Method**    | **MMLU ↑**   | **GSM8K ↑** | **HumanEval ↑** | **MMLU ↑**        | **GSM8K ↑** | **HumanEval ↑** |
> > | Non-watermark | 0.650      | 0.821     | 0.400         | 0.601           | 0.808     | 0.348         |
> > | KGW           | 0.567      | 0.726     | 0.104         | 0.536           | 0.582     | 0.092         |
> > | PATTERN-MARK  | 0.579      | 0.670     | 0.152         | 0.540           | 0.513     | 0.079         |
> > | dMARK         | **0.649**      | **0.814**     | **0.317**         | **0.596**           | **0.759**     | **0.201**         |
> > | dMARK + 3beam | **0.649**      | 0.774     | 0.207         | 0.588           | 0.723     | 0.134         |
> >
> > - DREAM
> >
> > |               | **Greedy** |           |               | **Multinomial** |           |               |
> > |---------------|:----------:|:---------:|:-------------:|:---------------:|:---------:|:-------------:|
> > | **Method**    | **MMLU ↑**   | **GSM8K ↑** | **HumanEval ↑** | **MMLU ↑**        | **GSM8K ↑** | **HumanEval ↑** |
> > | Non-watermark | 0.700      | 0.800     | 0.427         | 0.630           | 0.789     | 0.420         |
> > | KGW           | 0.558      | 0.661     | 0.287         | 0.523           | 0.444     | 0.134         |
> > | PATTERN-MARK  | 0.594      | 0.652     | 0.335         | 0.551           | 0.639     | 0.287         |
> > | dMARK         | **0.695**      | **0.746**     | **0.470**         | **0.647**           | **0.686**     | **0.390**         |
> > | dMARK + 3beam | **0.695**      | 0.701     | 0.342         | 0.636           | 0.648     | 0.262         |
> >
> > We have clarified this point in the revision by adding the corresponding experimental results to **Table 3** in **Section 4**.

---

> > ### Comment · Reviewer_mPfw · 2025-11-26
> >
> > Thank the authors for their rebuttal! After reading your rebuttal carefully, I find some points still confusing, including but not limited to:
> >
> > 1. "No probability biasing or rejection: The method of [1] increases the probability of green tokens (or rejects others). dMARK does not modify or bias token probabilities at any step. The model distribution is preserved exactly." and "In dMARK, watermarking is introduced solely through the decoding order, not probability adjustment. For every step, we use the model’s conditional probabilities exactly as-is." I don't quite think so. For a sanity check, if the joint distribution is preserved exactly after watermarking, then it is totally undistinguishable from the original distribution from information theory, and thus it's impossible to identify the watermarking. Apart from that sanity check, your arguments are wrong: Everytime one new token is sampled (no matter where the position is), you prefer the tokens in the green list, and only if the token is in the green list will it be decoded. You cannot argue that you preserve the distribution $p(y_i \mid y_{\mathcal{I}}, x)$ because any token outside the green list is filtered out. In that light, your rebuttal to W2 and Q1 is not correct.
> >
> > 2. "Detection rule: Our parity-based detection does not depend on previous tokens or n-grams, unlike the context-dependent hashing in [1]." I think the authors may have misunderstood my point. Kirchenbauer et al., 2023 has provided a self-hashing algorithm and it's Algorithm 3 in their paper. You have rebuttaled against a wrong algorithm. Please read my review carefully.
> >
> > Thank the authros again for the rebuttal. However, I find it unsatisfying and will remain my score of rejection.

---

> > > ### Author Response · Authors · 2025-11-28
> > > **Additional Response to Reviewer mPfw**
> > >
> > > Thank you again for the thoughtful and helpful feedback.
> > >
> > > ---
> > > 1\. We appreciate the reviewer’s careful reasoning and agree that our earlier wording (“the model distribution is preserved exactly”) was too strong and potentially misleading.
> > >
> > > Our intended claim is not that the effective joint distribution under dMARK is identical to the original model in practice, which would indeed make detection impossible, but rather that, unlike prior watermarking methods, we do not explicitly modify or reweight the token probabilities output by the model. Concretely, at every step we query the original $p_{\theta}(y_j|y_{\mathcal{I}}, x)$ and never rescale or renormalize these probabilities to boost “green” tokens, as in Kirchenbauer et al. (2023) and follow-ups.
> > >
> > > dMARK instead uses the degree of freedom available in order-agnostic decoding for dLLMs: we change which position to decode next, not the probabilities for each token at that position. In the idealized setting (order-agnostic models with exact conditionals and ancestral sampling) considered in the seminal work [1], any permutation of decoding order would indeed lead to the same joint distribution $p_{\theta}(y|x)$. In that sense, the theoretical model is “unbiased” with respect to decoding order.
> > >
> > > In practice, however, two factors break this invariance and create a detectable signal:
> > > - Imperfect learning: Real dLLMs only approximate the ideal conditional family. Different decoding orders interact differently with approximation error, so the realized joint distribution under dMARK does differ slightly from the baseline, even though we never explicitly change $p_{\theta}$
> > > - Heuristic decoding (greedy/multinomial): We do not perform exact ancestral sampling; we use greedy or multinomial sampling with a particular ordering. As discussed in [1], changing the order under such heuristics induces systematic biases that can be exploited for watermarking.
> > >
> > > Crucially, we do not filter or reject tokens outside $\mathcal{G}$ in the way self-hashing methods do. Tokens in $\mathcal{R}$ are still sampled from the same $p_{\theta}$; they are simply decoded later in the sequence, not removed from consideration. The watermark signal arises from how this parity-guided ordering interacts with imperfect conditionals and greedy/multinomial sampling, not from explicit probability boosting or rejection.
> > >
> > > In the revised manuscript, we removed phrases such as “preserved exactly’’ or “unbiasedness’’ and instead state the more precise claim of “no explicit probability reweighting.” We also added a brief discussion connecting our argument to the order-agnostic decoding analysis of Kim et al. (2025), explaining that while the joint distribution is invariant to decoding order in the idealized limit, in practice the watermark signal arises from model approximation error together with greedy/multinomial decoding heuristics.
> > >
> > > [1] Kim et al, Train for the Worst, Plan for the Best: Understanding Token Ordering in Masked Diffusions, ICML, 2025
> > >
> > > ---
> > > 2\. Thank you for the clarification and we apologize for the confusion. To respond more precisely, we explicitly compare dMARK with Algorithm 3 (self-hashing) from Kirchenbauer et al. (2023) and clarify the differences. In particular, we highlight two distinctions:
> > > - Token Substitution vs. Order Guidance
> > >
> > > In Algorithm 3, if the model’s sampled choice falls in the red list, the algorithm examines lower-ranked alternatives in sequence, searching for a "green" token within a $\delta$ margin. The original choice is replaced with the first "green" token found, or retained if none lies within the threshold. This substitution directly modifies the token that the model would have otherwise produced.
> > > In contrast, dMARK does not modify or intervene in the model’s token selection. The token chosen by the model is always preserved; it does not reject, re-rank, or replace the selected token. Instead, it only influences the order in which masked positions are generated. In other words, dMARK does not suppress unmatched tokens; it simply defers filling those positions until all matched tokens have been generated.
> > > - Context-Dependent vs. Context-Independent
> > >
> > > In Algorithm 3, self-hashing takes pairs of the current candidate token $s^{(t)}$ and past tokens $s^{(t-i)}$ (within a sliding window) as PRF inputs. The index $i^\star$ yielding the minimal PRF output is selected, and $(s^{(t)}, s^{(t-i^\star)})$ determines whether the candidate is green or red. Thus, the watermark decision at position $t$ depends on a specific token in the preceding context.
> > > In contrast, dMARK does not rely on past tokens to classify candidates as matched or unmatched. It uses a context-independent mechanism: for each masked position, a deterministic function assigns each candidate token a binary value. Tokens whose values match the position’s parity are treated as “matched”, and otherwise as “unmatched”. This partitions the vocabulary per position, independent of previously generated content.

---

### Meta-Review · Area_Chair_wxWn · 2026-01-07

**Summary:**

Reviewers appreciated that the paper introduces a watermarking method tailored to discrete diffusion language models (dLLMs). In particular, they noted that the self-hashing style procedure is easy to implement (mPfw), that the proposed approach can be combined with different decoding strategies and is well motivated (DGd2), and that the experiments study trade-offs among detectability, text quality, and robustness. At the same time, reviewers raised several limitations, especially around missed or unclear connections to prior work, and two of the three reviewers explicitly stated in the discussion that they would maintain their pre-rebuttal (reject-leaning) stance after reading the rebuttal and engaging with the authors.

**Reviewer Concerns:**

Reviewer mPfw emphasized that the core “self-hashing” idea is not new, arguing it is closely related to prior self-hashing watermarking (specifically pointing to Kirchenbauer et al. self-hashing algorithm). They also questioned the rigor and framing of the paper’s “unbiasedness” claim. The reviewer noted that text quality appeared to degrade substantially under watermarking. After the rebuttal, mPfw posted that they still had points of confusion and continued to disagree with the authors’ characterization of unbiasedness, and they reiterated that the closest comparison should be to the self-hashing algorithm they referenced (Algorithm 3). They concluded that the rebuttal was unsatisfying and stated they would maintain their original reject score.

Reviewer DGd2 found the motivation and generality across decoding strategies promissing, but asked for stronger robustness evaluation (including more sophisticated transformations), clearer comparisons to existing watermarking methods, and additional analysis of computational overhead. In rebuttal, the authors responded with added robustness evaluation (including paraphrasing), added comparisons, and overhead measurements, along with additional clarifying material. Alas, the reviewer did not seem to engage further.

Reviewer JtBe raised concerns about presentation and clarity, questioned novelty because the method is assembled from prior components, and flagged missing Ethics and Reproducibility statements in the original manuscript. In the post-rebuttal discussion, JtBe acknowledged the revisions but stated there remained considerable room for improvement and that they still had reservations about novelty, maintaining their original score.

**Reviewer Scores:**

Reviewer mPfw andJtBe explicitly noted not change in score. Reviewer DGd2 did not engage but, given their initial review, it is unlikely that their score would increase substantially.

---

### Decision · Program_Chairs · 2026-01-26

Reject